# Impaired lysosomal acidification triggers iron deficiency and inflammation in vivo

King Faisal Yambire[1], Christine Rostosky[2], Takashi Watanabe[3],
David Pacheu-Grau[1], Sylvia Torres-Odio[4], Angela Sanchez-Guerrero[1,2],
Ola Senderovich[5], Esther G Meyron-Holtz[5], Ira Milosevic[2], Jens Frahm[3],
A Phillip West[4], Nuno Raimundo[1]*

[1]Institute of Cellular Biochemistry, University Medical Center Goettingen, Goettingen, Germany; [2]European Neuroscience Institute, a Joint Initiative of the Max-Planck Institute and of the University Medical Center Goettingen, Goettingen, Germany; [3]Biomedizinische NMR, Max-Planck Institute for Biophysical Chemistry, Goettingen, Germany; [4]Department of Microbial Pathogenesis and Immunology, Texas A&M University Health Science Center, Austin, United States; [5]Faculty of Biotechnology and Food Engineering, Technion Israel Institute of Technology, Haifa, Israel

**\*For correspondence:**
nuno.raimundo@med.uni-goettingen.de

**Competing interests:** The authors declare that no competing interests exist.

**Abstract** Lysosomal acidification is a key feature of healthy cells. Inability to maintain lysosomal acidic pH is associated with aging and neurodegenerative diseases. However, the mechanisms elicited by impaired lysosomal acidification remain poorly understood. We show here that inhibition of lysosomal acidification triggers cellular iron deficiency, which results in impaired mitochondrial function and non-apoptotic cell death. These effects are recovered by supplying iron via a lysosome-independent pathway. Notably, iron deficiency is sufficient to trigger inflammatory signaling in cultured primary neurons. Using a mouse model of impaired lysosomal acidification, we observed a robust iron deficiency response in the brain, verified by in vivo magnetic resonance imaging. Furthermore, the brains of these mice present a pervasive inflammatory signature associated with instability of mitochondrial DNA (mtDNA), both corrected by supplementation of the mice diet with iron. Our results highlight a novel mechanism linking impaired lysosomal acidification, mitochondrial malfunction and inflammation in vivo.

## Introduction

Lysosomal function is now recognized as a key factor in cellular and tissue health. Recessive mutations in genes encoding lysosomal proteins result in over 50 severe lysosomal storage diseases, and carriers of these mutations are at risk of many neurodegenerative diseases, such as Parkinson's (*Ramirez et al., 2006*; *Murphy et al., 2014*; *Nguyen et al., 2019*), Alzheimer's (*Lee et al., 2010*), amyotrophic lateral sclerosis (*Corrionero and Horvitz, 2018*), frontotemporal lobar degeneration (*Lie and Nixon, 2019*), among others (*Fassio et al., 2018*).

The lysosomes are now recognized as key players in cellular signaling and nutrient sensing, in addition to their roles as terminal platform of autophagy and endocytosis (*Perera and Zoncu, 2016*). Lysosomes are also involved in the intracellular partitioning of several cellular building blocks, such as amino acids, cholesterol and sphingomyelin, as well as metals including calcium (Ca) and iron (Fe) (*Lim and Zoncu, 2016*).

There are different populations of lysosomes in each cell, which can be distinguished by their position, size, acidification and reformation properties (*Bright et al., 2016*). Yet, most lysosomal functions rely on the acidification of the lysosomal lumen, as the vast majority of the enzymes residing in the lysosome have an optimal function in the pH range of 4.5–5 (*Perera and Zoncu, 2016*).

The acidic pH of the lysosomal lumen is the result of an electrochemical gradient maintained mostly by the vacuolar ATPase (v-ATPase) (*Wang and Semenza, 1993*), with contribution from the chloride channel CLC-7 (*Mindell, 2012*). The v-ATPase is a multisubunit complex with two domains, the membrane-associated $V_O$ and the soluble hydrolytic $V_1$, which hydrolyzes ATP to pump protons to the lysosomal lumen against a concentration gradient (*Stransky et al., 2016*). The existence of specific v-ATPase inhibitors such as bafilomycin (Baf; interacts with the $V_O$ ring, inhibiting proton translocation) and saliphenylhalamide (saliphe; locks v-ATPase in an assembled state) provides pharmacological tools to assess loss of v-ATPase activity (*Bowman et al., 1988*; *Garcia-Rodriguez et al., 2015*; *Xie et al., 2004*).The v-ATPase exists also in other cellular organelles, such as endosomes, Golgi complex and secretory vesicles (*Farsi et al., 2018*), but the effects of v-ATPase loss-of-function most severely relate to its role in lysosomal acidification (*Lie and Nixon, 2019*).

Decreased activity of the v-ATPase has been linked to age-related decrease in lysosomal function and neurodegenerative diseases (*Lee et al., 2010*; *Nixon, 2013*; *Korvatska et al., 2013*; *Bagh et al., 2017*; *Fassio et al., 2018*). It has also been shown that impaired acidification of the yeast vacuole, the evolutionary ancestor of lysosomes, results in decreased mitochondrial function and accelerated aging (*Hughes and Gottschling, 2012*). Nevertheless, the mechanisms by which impaired lysosomal acidification result in aging and disease remain poorly characterized. In addition, the v-ATPase has been recognized as a therapeutic target in cancer, given that its inhibition positively correlates with decreased tumor mass (*McGuire et al., 2016*). Therefore, it is pivotal to characterize the mechanisms underlying the cellular response to v-ATPase inhibition.

Importantly, the potent inhibition of v-ATPase causes cell death, but the underlying mechanisms also remain unclear. The understanding of which signaling pathways are elicited in response to v-ATPase inhibition would allow the definition of therapeutic targets, for example for neurodegenerative diseases and for cancer.

Here, we show that inhibition of the v-ATPase results in impairment of lysosomal iron metabolism, which causes iron deficiency in the cytoplasm and in mitochondria. This results in activation of the pseudo-hypoxia response, loss of mitochondrial function and cell death. These effects were all ablated by iron repletion in a form that can be imported across the plasma membrane, independently of the lysosomes. Notably, iron deficiency is sufficient to trigger inflammatory signaling in cultured neurons as well as in vivo. A mouse model of impaired lysosomal acidification shows iron deficiency, activation of the pseudo-hypoxia response, and pervasive inflammation, all detectable long before the disease onset. All these phenotypes could be rescued by increasing the levels of iron in the animal diet.

## Results

### v-ATPase inhibition triggers hypoxia-inducible factor-mediated response

With the goal of identifying signaling events caused by impaired organelle acidification, we analyzed several transcriptome datasets of cells treated with the v-ATPase inhibitor bafilomycin. These datasets include bafilomycin treatment of HeLa cells (GSE16870) (*Straud et al., 2010*), colon carcinoma cells (GSE47836) (*Dürrbaum et al., 2014*) and retinal pigment epithelial cells (GSE60570) (*Santaguida et al., 2015*). We performed multi-dimensional transcriptome analyses in these datasets, aiming at the identification of signaling pathways, networks and transcription factors (*Murdoch et al., 2016*; *Raimundo et al., 2012*; *Raimundo et al., 2009*; *Schroeder et al., 2013*; *Tyynismaa et al., 2010*; *West et al., 2015*; *Yambire et al., 2019*). We reasoned that those transcription factors (TF) showing similar behavior in the three datasets of bafilomycin-treated cells would be the main regulators of the response to loss of acidification, independently of the cell type. Therefore, we crossed the TF list associated with each dataset, to determine which of those were involved in all three datasets, and found eight common TF, of which seven were predicted as active and one as repressed (*Figure 1A*). These TF are associated with autophagy (NUPR1), cholesterol homeostasis (SREBF1, SREBF2), hypoxia response (HIF-1α and EPAS1, which is also known as HIF-2α) and diverse stress responses (p53, myc, FoxO3a), and form a highly interconnected network (*Figure 1B*). To determine which biological processes were associated with these TF, and identify which of them

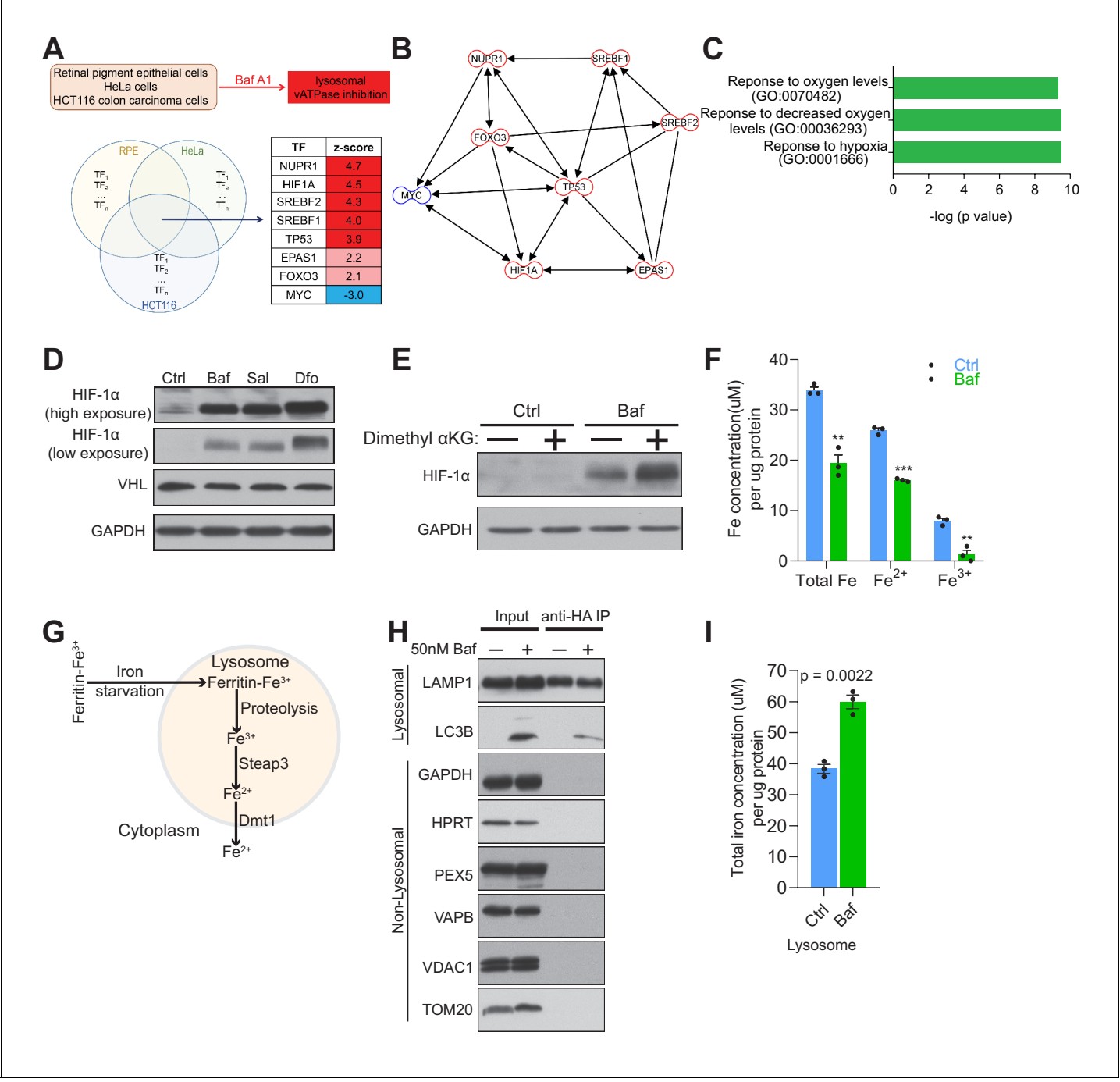

**Figure 1.** Lysosomal v-ATPase inhibition triggers HIF-1α activation. (**A**) Venn diagram illustrating common significantly changed upstream regulators (shown as a table) mediating differential gene expression in bafilomycin (Baf)-treated cells in three different transcriptome datasets from HeLa, HCT116 and RPE cells. (**B**) Transcription factor network of the significantly changed upstream regulators found in *Figure 1A*. (**C**) Metascape gene set enrichment analyses show significant overrepresentation of the transcription factors (*Figure 1A*) in the hypoxia response. (**D**) Immunoblot of HIF-1α and VHL with GAPDH as loading control in cell lysates of 500 nM Baf- and 500 nM saliphe (Sal)-treated fibroblasts. 300 µM Dfo is used as a positive control for the iron deficiency response. (**E**) Western blot of whole cell extracts showing HIF-1α levels in control and 500 nM Baf-treated fibroblasts supplemented with cell-permeant αα-ketoglutarate (5 mM). GAPDH is used as a loading control (n = 3). (**F**) Total, ferrous and ferric iron concentrations in fibroblasts treated with or without 500 nM Baf for 24 hr. Results are summarized as mean ± SEM of three experimental measures shown as black dots. **p<0.01; ***p<0.001, unpaired two-tailed t-test. (**G**) Illustration of the intralysosomal pathway of iron homeostasis handling (**H**) Immunoblot of lysosomes following purification by the lysoIP method in 3x HA-tagged Tmem192 mouse embryonic fibroblasts treated with or without 50 nM Baf for 24 hr. Lysosomal purity is shown by enrichment of LAMP1 and LC3B in the lysosomal compartment, while excluding GAPDH and HPRT (cytosol), PEX5 (peroxisome), VAPB (endoplasmic reticulum), and VDAC1 and TOM20 (mitochondria), n = 3. (**I**) Total iron concentration in lysosomes purified from

*Figure 1 continued on next page*

*Figure 1 continued*

fibroblasts treated with or without 50 nM Baf for 24 hr. Results are shown as mean ± SEM of three experimental measures shown as black dots. The indicated p value is the unpaired two-tailed t-test with Welch's correction.

were most upstream, we performed a pathway analysis using Metascape, and found that the most affected processes dealt with cellular response to hypoxia (*Figure 1C*).

The major coordinator of the cellular response to hypoxia is the transcription factor hypoxia-inducible factor-1α (*Majmundar et al., 2010*), which is included in the list of TFs responding to bafilomycin. Furthermore, HIF-2α (EPAS1), whose activation mechanism is similar to HIF-1α, is also part of the TF list. The HIF transcription factors function as heterodimers of a regulated α-subunit and the constitutive β-subunit (*Majmundar et al., 2010*). The α subunits are regulated post-translationally by the prolyl hydroxylases. These enzymes are di-oxygenases of the α-ketoglutarate-dependent super-family, and hydroxylate HIF-1α and HIF-2α in the presence of $O_2$, $Fe^{2+}$ and α-ketoglutarate (*Majmundar et al., 2010*). Hydroxylated HIF-1α and HIF-2α are then recognized by the ubiquitin ligase VHL and targeted to the proteasome for degradation. Notably, the protein levels of HIF-1α were increased in bafilomycin (or Baf in figures)-treated fibroblasts (*Figure 1D*). A similar result was obtained in fibroblasts treated with a different inhibitor of the lysosomal v-ATPase, saliphenylhalamide (henceforth, saliphe [or Sal in figures]) (*Figure 1D*). It is known that iron chelation results in HIF-1α accumulation (*Wang and Semenza, 1993*); so, as positive control for the HIF-1α protein accumulation, we treated cells with an iron chelator, deferoxamine (Dfo in figures). Deferoxamine treatment resulted in the expected robust increase in HIF-1α protein levels (*Figure 1D*). Accumulation of HIF-1α can be caused by decreased function of the prolyl hydroxylases or decreased levels of $O_2$, $Fe^{2+}$, VHL or α-ketoglutarate (*Raimundo et al., 2011*). Because the fibroblasts used in this study are maintained under normoxic conditions, the lack of $O_2$ is not a factor. We therefore tested if VHL, α-ketoglutarate or $Fe^{2+}$ could explain the accumulation of HIF-1α and HIF-2α. The protein levels of VHL were not changed in bafilomycin-treated fibroblasts (*Figure 1D*). To test if the reason why HIF-1α was accumulating upon v-ATPase inhibition was due to decreased levels of α-ketoglutarate, we added a cell-permeant form of α-ketoglutarate to the bafilomycin-treated fibroblasts, and found no decrease in the accumulation of HIF-1α (*Figure 1E*). Then, we tested if there were any perturbations in Fe levels. We therefore assessed the total, $Fe^{2+}$ and $Fe^{3+}$ levels in bafilomycin-treated and control fibroblasts, and found a robust decrease in total, $Fe^{2+}$ and $Fe^{3+}$ (*Figure 1E*). The hydroxylation of HIF-1α, which leads to its ubiquitination and degradation, is $Fe^{2+}$-depedent, and therefore lower $Fe^{2+}$ levels are in agreement with the accumulation of HIF-1α. Altogether, these results suggest that perturbation of Fe homeostasis by v-ATPase inhibition triggers the pseudo-hypoxia HIF-mediated response, in agreement with a prior study (*Miles et al., 2017*).

When the cytoplasmic $Fe^{2+}$ levels are low, intracellular iron is mobilized by autophagy of ferritin (ferritinophagy) (*Mancias et al., 2014*) and mitochondria (mitophagy) (*Allen et al., 2013*; *Schiavi et al., 2015*), which deliver iron directly to the lysosomes (via autophagosomes). In parallel, transferrin receptor uptake first releases iron in the endosomes, which is then delivered by kiss-and-run to mitochondria (*Hamdi et al., 2016*) or released to the cytoplasm. In the endosomes/lysosomes, $Fe^{3+}$ is reduced to $Fe^{2+}$ by the enzyme STEAP3, and $Fe^{2+}$ (but not $Fe^{3+}$) is released to the cytoplasm by DMT1 (SLC11A2) or MCOLN1 (*Figure 1G*) (*Dong et al., 2008*; *Touret et al., 2003*). To test whether vATPase inhibition results in cytoplasmic $Fe^{2+}$ deficiency due to retention of iron in the lysosomes, we purified lysosomes from fibroblasts using an immunoprecipitation protocol (*Abu-Remaileh et al., 2017*). The lysosomes obtained from control and bafilomycin-treated fibroblasts showed enrichment in lysosomal proteins (e.g. LAMP1), while mitochondrial (VDAC1, TOM20), endoplasmic reticulum (VAPB), peroxisomal (PEX5) and cytoplasmic proteins (HPRT, GAPDH) were only detected in the input lanes (*Figure 1H*). Using these purified lysosomes, we measured their total iron content, and found a robust increase in lysosomal iron in the bafilomycin-treated cells (*Figure 1I*) in agreement with the retention of iron in lysosomes following impaired lysosomal acidification.

## Lysosomal vATPase inhibition causes functional iron deficiency

The cytoplasmic concentration of $Fe^{2+}$ is controlled by the iron regulatory protein 1 (IRP1), the IRP2 and the ferritinophagy receptor NCOA4 (*Mancias et al., 2014*; *Rouault, 2015*). When the levels of cytoplasmic $Fe^{2+}$ are down, IRP activity mediates a reduction in the expression of the iron storage protein ferritin, and an increase in transferrin receptor leading to extracellular (transferrin-bound) iron uptake (*Rouault, 2015*).

We first tested if the lower cellular $Fe^{2+}$ levels when the v-ATPase is inhibited can affect iron homeostasis. We monitored cellular iron homeostasis by measuring the protein levels of ferritin light (FTL1) and heavy chain (FTH1) subunits and the transcript levels of transferrin receptor (TFRC). The lysosomal chelator deferoxamine, was used as positive control given that it retains iron in the lysosome and thus impedes its release to the cytoplasm, thus triggering functional iron deficiency (*Doulias et al., 2003*; *Kurz et al., 2006*). As TFRC mRNA has a 3' iron responsive element, its transcript levels are expected to increase upon iron deficiency. We observed in fibroblasts treated with the v-ATPase inhibitors, bafilomycin and saliphe, increased *Tfrc* transcript levels and decreased protein amounts of FTL1 and FTH1 (*Figure 2*). As expected, deferoxamine treatment resulted in increased *Tfrc* transcript (*Figure 2—figure supplement 1A*) and decreased protein levels of the ferritin light and heavy chain subunits, FTL1 and FTH1 (*Figure 2A*). Importantly, the treatment with Deferoxamine does not impact lysosomal function, as assessed by the protein levels of autophagy markers SQSTM1 and LC3B-II (*Figure 2—figure supplement 1B*). Furthermore, we stained fibroblasts with the probe FerroOrange, which reacts specifically with $Fe^{2+}$ but not $Fe^{3+}$, and found that the signal intensity of FerroOrange was sharply decreased in bafilomycin-treated cells (*Figure 2B*). Notably, iron deficiency elicited by bafilomycin is not a consequence of the concentration employed, as lower concentration yields a similar result (*Figure 2—figure supplement 1C*). Similarly, chloroquine, a lysosomotrophic weak base, which perturbs lysosomal acidification by dissipating the pH gradient across the lysosomal membrane but not by inhibiting the vATPase, also results in functional iron deficiency (*Figure 2—figure supplement 1C*).

Because the iron-deficiency response caused by v-ATPase inhibition is due to retention of iron in the lysosome, we supplemented the growth medium with Fe-citrate, which allows iron to be imported through transporters in the plasma membrane (*Ofer et al., 1981*), thus bypassing the endolysosomal pathway. When fibroblasts are treated simultaneously with v-ATPase inhibitors and Fe-citrate, the iron-deficiency response is deactivated, as assessed by the transcript levels of *Tfrc* (*Figure 2C*). To ensure that his effect was specifically due to Fe, Na-citrate was used as control. This result underscores that the iron homeostatic response is regulated by the cytoplasmic iron levels (*Rouault, 2015*), and that retention of iron in the lysosome triggers cytoplasmic iron deficiency, which can be resolved by iron import independently of the endolysosomal pathway.

To directly test if the accumulation of HIF-1α upon v-ATPase inhibition was due to the perturbation in Fe homeostasis, we supplemented the growth medium of the bafilomycin- or saliphe-treated fibroblasts with Fe-citrate, which restored HIF-1α to the barely detectable amounts observed in control cells (*Figure 2D*). The striking normalization of ferritin levels upon Fe-citrate supplementation of bafilomycin-treated fibroblasts further underscores that Fe-citrate is being taken up by the cells and resolves the iron deficiency (*Figure 2D*).

To further probe the impact of HIF-1α accumulation upon v-ATPase inhibition, we tested if the transcriptional targets of HIF-1α were affected. We observed that the transcript levels of HIF-1α targets are robustly induced in bafilomycin- and saliphe-treated fibroblasts, and are restored to basal levels when the cells are co-treated with v-ATPase inhibitors and Fe-citrate (*Figure 2E* and *Figure 2—figure supplement 1D*).

We then sought to test whether genetic perturbation of lysosomal acidification also triggers iron deficiency and impacts HIF accumulation. We silenced the expression of *Atp6v1h*, encoding an essential subunit of the vATPase, using siRNAs (*Figure 2—figure supplement 1E*). The *Atp6v1h*-silenced fibroblasts show impaired lysosomal acidification (*Figure 2—figure supplement 1F and G*). In the *Atp6v1h*-silenced fibroblasts we observed increased expression of *Tfrc*, decreased protein levels of FTH1 and FTL1, and accumulation of HIF-1α (*Figure 2—figure supplement 1E*), as well as increased transcript levels of most HIF-1α target genes (*Figure 2—figure supplement 1H*). Therefore, inhibition of lysosomal acidification using a genetic approach also results in functional iron deficiency and HIF-1α activation.

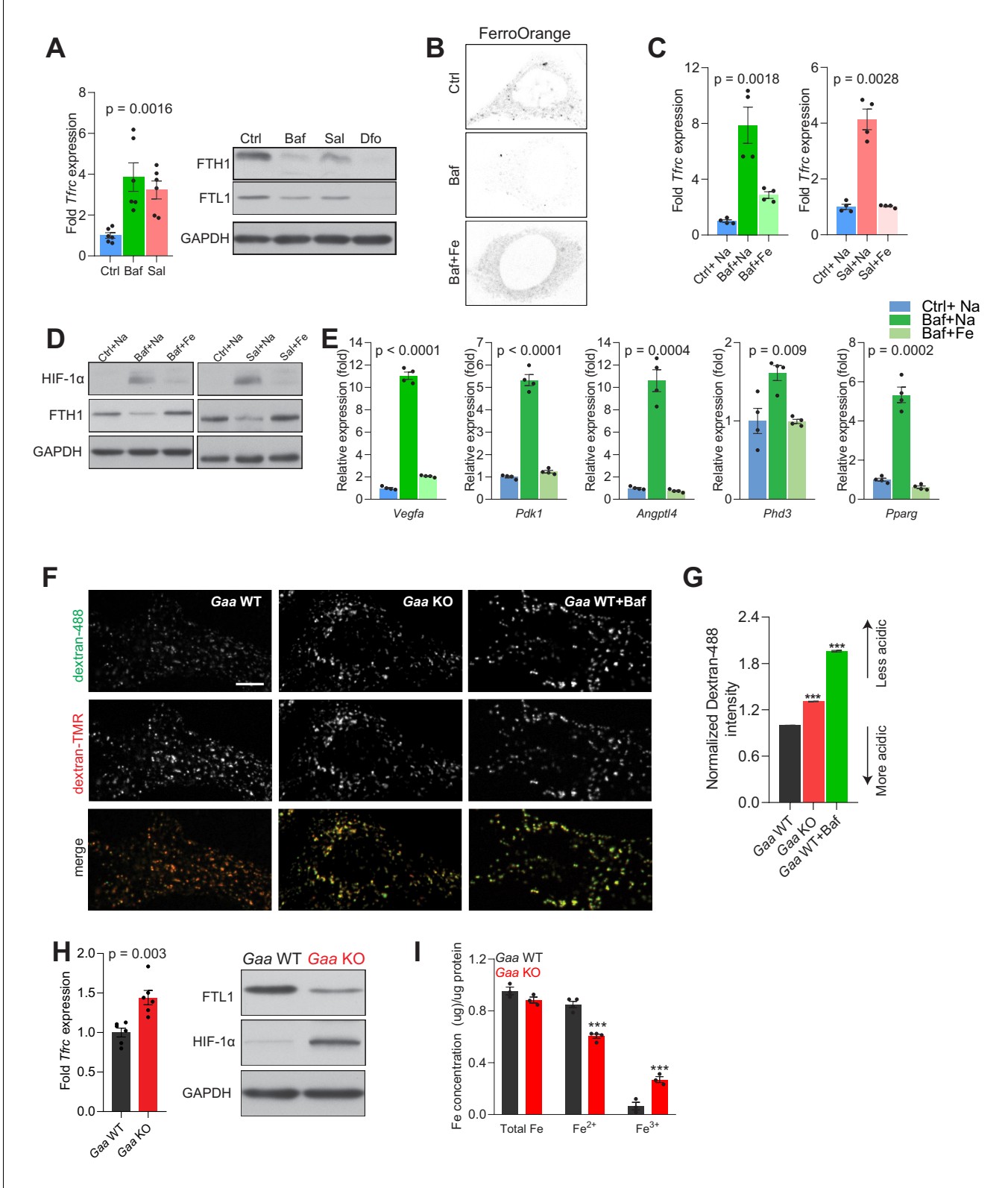

**Figure 2.** Lysosomal iron efflux regulates HIF-1α response. (**A**) Increased *Tfrc* transcript levels in 500 nM Baf- and 500 nM Sal-treated mouse embryonic fibroblasts relative to untreated fibroblasts. Western blot showing decreased FTH1 and FTL1 protein levels in Baf- and Sal-treated fibroblasts (n = 6). GAPDH is used as loading control. *Tfrc* expression is depicted as bars representing mean ± SEM, n = 6; shown as black dots. p value is determined by the Welch's one-way ANOVA as differences between untreated group, and Baf- and Sal-treatments. (**B**) Representative images of FerroOrange staining

*Figure 2 continued on next page*

*Figure 2 continued*

of cytoplasmic labile iron pools in control, 500 nM Baf-treated and 500 nM Baf-treated fibroblasts with 150 µM iron citrate supplementation. Note the reduced staining in Baf-treated fibroblasts relative to the other conditions. (C) mRNA levels of *Tfrc* in control, 500 nM Baf-treated and 500 nM Baf-treated fibroblasts with 150 µM iron citrate supplementation (left) or in control, 500 nM Sal-treated and 500 nM Sal-treated fibroblasts with 150 µM iron citrate supplementation (right) for 24 hr. Bar graphs depict mean ± SEM of four independent experimental measures (shown as black dots). p values represent Welch's one-way ANOVA with Dunnett's correction for multiple comparisons, estimated as differences between Baf- or Sal-treated cells and other experimental groups. (D) Whole cell immunoblots of HIF-1α and FTH1 in fibroblasts treated with 500 nM Baf or 500 nM Baf + 150 µM iron citrate (left) or with 500 nM Sal and 500 nM Sal + 150 µM iron citrate (n = 4). GAPDH is used as loading control. (E) Transcript levels of HIF-1α target genes in fibroblasts treated with 500 nM Baf or with 500 nM Baf + 150 µM iron citrate. The mean ± SEM of four biological replicates (black dots) is shown. p-values are determined by Welch's one-way ANOVA with Dunnett's correction for multiple comparisons (all experimental groups compared to Baf-treated cells). (F–G) *Gaa* KO fibroblasts display impaired lysosomal acidification. Representative spinning-disk microscopy images for *Gaa* WT and *Gaa* KO MEFs co-stained with Dextran-Oregon Green 488 and Dextran-TMRM are shown with Baftreatment in *Gaa* WT used as a positive control for impaired acidification. Scale bar 2 µm. (G) Quantification of the intensity of Dextran-Oregon Green in Dextran-TMRM positive punta shows increased Dextran-Oregon Green intensity in Gaa KO fibroblasts and in Baf-treated *Gaa* WT fibroblasts. Bar graphs depict mean ± SEM, n = 3 independent experiments, with 50 cells per condition from each experiment. \*\*\*$p<0.001$, determined by the unpaired two-tailed t-test with Welch's correction (H) Increased *Tfrc* expression (left) in $Gaa^{-/-}$ fibroblasts (n = 6, depicted as black dots). p value is determined by the unpaired two-tailed t-test with Welch's correction. Whole cell immunoblots of FTL1 and HIF-1α (right) in $Gaa^{-/-}$ fibroblasts (n = 6). GAPDH is used as loading control. (I) Total, ferrous and ferric iron concentrations in fibroblasts prepared from $Gaa^{-/-}$ and their wild type littermate controls. Results are summarized as mean ± SEM of experimental measures shown as black dots. Differences between means shown as actual p values are determined by the unpaired two-tailed t-test with Welch's correction.

The online version of this article includes the following figure supplement(s) for figure 2:

**Figure supplement 1.** HIF-1a mediated response is triggered by lack of lysosomal iron efflux.

To test whether other lysosomal defects which impair lysosomal acidification would similarly result in functional iron deficiency, we used fibroblasts derived from mice lacking acid α-glucosidase (*Gaa*, mice referred henceforth as *Gaa*-KO), a lysosomal enzyme mutated in Pompe's disease. The immediate biological consequence of the loss of GAA is lysosomal glycogen storage (*Raben et al., 1998*). Furthermore, it was reported that a large portion of lysosomes is not able to acidify in *Gaa*-KO fibroblasts (*Fukuda et al., 2006*). We assessed lysosomal pH in the *Gaa*-KO fibroblasts (*Figure 2F*) and observed impaired acidification, albeit less robust than observed when inhibiting the vATPase or silencing *Atp6v1h* (*Figure 2F–G*). The *Gaa*-KO fibroblasts present iron deficiency as shown by increased *Tfrc* transcript level, and decreased FTL1 levels (*Figure 2H*). Notably, these fibroblasts also present HIF-1α accumulation (*Figure 2H*). Furthermore, the *Gaa*-KO fibroblasts show a decrease in $Fe^{2+}$ (*Figure 2I*), in agreement with the iron deficiency response and the accumulation of HIF-1α. The increase in $Fe^{3+}$ suggests a defect in the endolysosomal reduction of $Fe^{3+}$ to $Fe^{2+}$.

Altogether, these results show that v-ATPase inhibition results in decreased cytoplasmic $Fe^{2+}$, which impairs the hydroxylation-mediated degradation of HIF-1α and culminates in the accumulation and activation of this transcription factor.

## Mitochondrial biogenesis and function require iron

Having shown that endolysosomal acidification is required for $Fe^{2+}$ release from endo/lysosomes to the cytoplasm, we sought to test if mitochondrial iron content was also affected. We loaded the cells with the probe Mito-FerroGreen, which localizes to mitochondria and reacts specifically with $Fe^{2+}$. The cells treated with bafilomycin showed a robust decrease in Mito-FerroGreen signal (*Figure 3A*). This effect was ablated in the presence of Fe-citrate (*Figure 3A*). This result shows that supply of iron to mitochondria is also impaired when the v-ATPase is inhibited, and that mitochondria seem to be able to uptake iron from cytoplasm when the endolysosomal pathway is ineffective.

Mitochondria are a key component of Fe-S cluster synthesis, which is essential for the proper function of the respiratory chain. To assess if mitochondrial function was impacted by the decrease in $Fe^{2+}$, we performed real-time respirometry, which monitors $O_2$ consumption in real-time as a proxy for mitochondrial respiratory chain activity. We observed that fibroblasts treated with either v-ATPase inhibitor bafilomycin or saliphe showed a robust decrease in mitochondrial $O_2$ consumption (*Figure 3—figure supplement 1A*). A similar result was obtained when perturbing lysosomal acidification using a lower concentration of bafilomycin, or using chloroquine (*Figure 3—figure supplement 1B*). Interestingly, treatment of fibroblasts with the lysosomal iron chelator deferoxamine resulted in virtual absence of mitochondrial respiratory chain activity (*Figure 3—figure supplement*

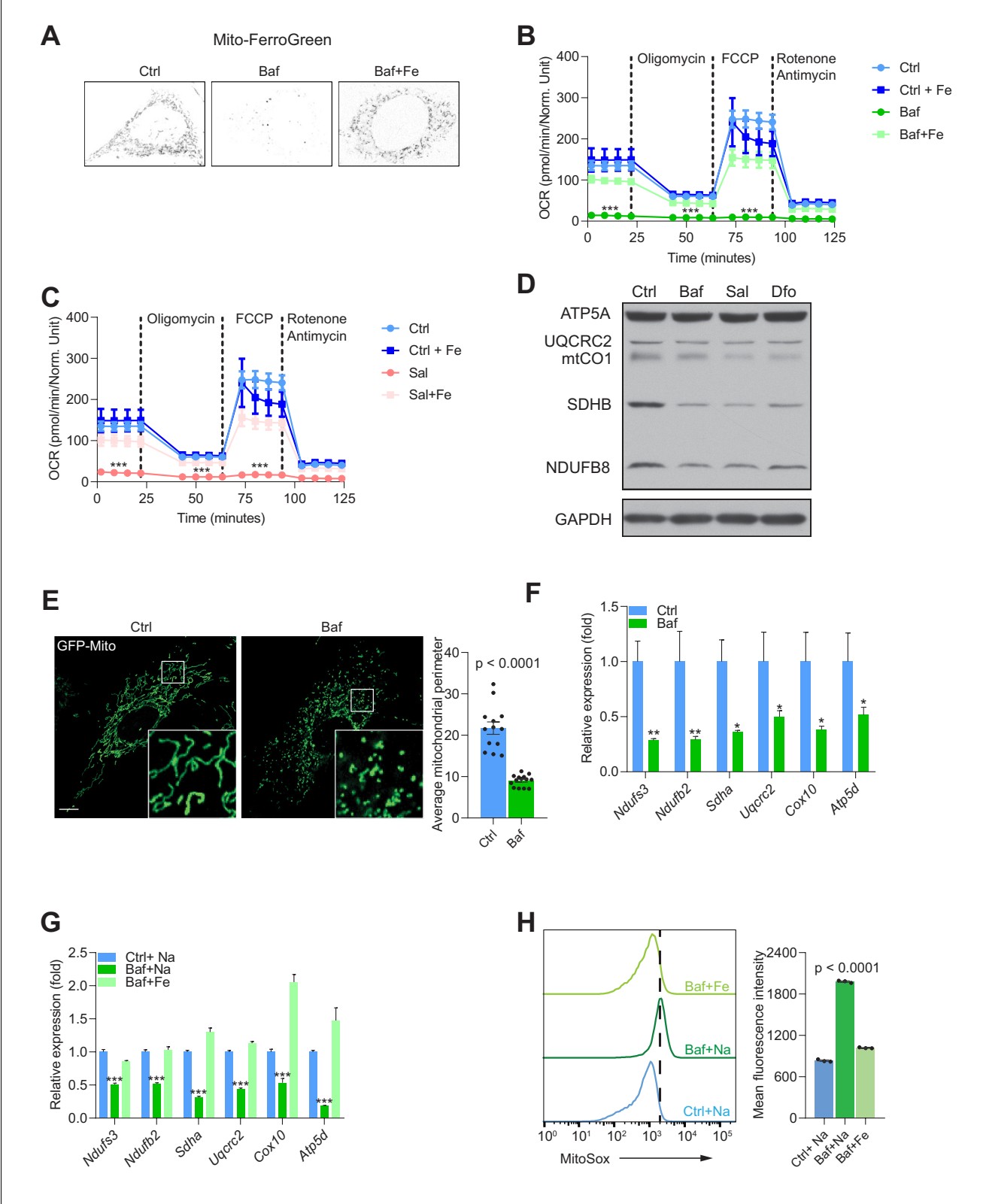

**Figure 3.** Mitochondrial biogenesis and function are dependent on endolysosomal iron supply. (**A**) Representative image Mito-FerroGreen staining of mitochondrial labile iron pools in control, 500 nM Baf- and 500 nM Baf + 150 µM Fe-citrate treated fibroblasts. Note the barely detectable labile iron levels in mitochondria of Baf-treated fibroblasts. (**B–C**) Mitochondrial oxygen consumption rates (OCR) in 500 nM Baf- and 500 nM Baf-treated fibroblasts with 150 µM iron citrate supplementation (**B**) or in 500 nM Sal- and 500 nM Sal-treated fibroblasts with 150 µM iron citrate supplementation

*Figure 3 continued on next page*

*Figure 3 continued*

(**C**). Results represent mean ± SEM, three independent experimental replicates. Each experimental replicate is calculated from the average of 8 technical replicates. \*\*\*p<0.001, Welch's one-way ANOVA with Dunnett's correction for multiple comparison (all experimental groups compared to Baf- or Sal-treated cells). (**D**) Western blot of whole cell extracts showing ATP5A, UQCRC2, mtCO1, SDHB and NDUFSB in control, 500 nM Baf- and 500 nM Sal-treated fibroblasts (n = 6). 300 μM Dfo is used as positive control for iron deficiency. Note the reduction of mitochondrial proteins in treated cells relative to controls. GAPDH is used as loading control. (**E**) Representative images of fibroblasts transfected with GFP-Mito and treated with 500 nM Baf for 24 hr to show the mitochondrial network. Note the prevalent mitochondrial fragmentation in Baf-treated fibroblasts. Number of cells is shown as black dots per condition in bars representing mean ± SEM of mitochondrial perimeter from three independent experiments. p value is estimated by the Mann-Whitney U-test. Scale bar, 10 μm. (**F**) Transcript levels of nuclear-encoded mitochondrial genes in 500 nM Baf-treated fibroblasts. Bars depict mean ± SEM of six independent experimental measures. \*p<0.05; \*\*p<0.01; \*\*\*p<0.001, unpaired two-tailed t-test with Welch's correction. (**G**) Transcript levels of nuclear-encoded mitochondrial genes in 500 nM Baf-treated fibroblasts and 500 nM Baf-treated fibroblasts with150μM iron citrate supplementation. Results are shown as mean ± SEM, n = 8. \*p<0.05; \*\*p<0.01; \*\*\*p<0.001, Welch's one-way ANOVA with multiple test corrections made by the Dunnett's method. All comparisons were made between Baf-treated cells and other experimental conditions. (**H**) Mitochondrial superoxide levels in 500 nM Baf- and 500 nM Baf-treated fibroblasts with 150 μM iron citrate supplementation. Differences are depicted as mean fluorescent intensities of the superoxide-sensitive and mitochondrial-targeted dye, Mitosox. Error bars represent SEM of three independent experimental measures (black dots). p value is Welch's one-way ANOVA with Dunnett's correction for multiple corrections (all conditions compared to Baf-treated cells).

The online version of this article includes the following figure supplement(s) for figure 3:

**Figure supplement 1.** Lysosomal iron efflux is essential for mitochondrial biogenesis and function.

**Figure supplement 2.** Impaired lysosomal iron efflux results in decreased mitochondrial biogenesis and function.

*1C*). Notably, supplementation of the medium with Fe-citrate was sufficient to return mitochondrial respiratory chain activity to normal levels in cells treated with bafilomycin (*Figure 3B*), saliphe (*Figure 3C*) or with deferoxamine (*Figure 3—figure supplement 1D*).

We then explored what may be driving the decrease in mitochondrial respiration. Multiple causes are possible, including a decrease in the mass or the quality of the mitochondrial network, or a metabolic shift that shunts pyruvate away from aerobic metabolism. First, we assessed the protein levels of respiratory chain subunits in whole cell extracts by Western blot, and observed that they are decreased when v-ATPase is inhibited (*Figure 3D*). Similar results were observed in fibroblasts treated with the iron chelator deferoxamine (*Figure 3D*), chloroquine or lower concentration of bafilomycin (*Figure 3—figure supplement 1F*). To assess mitochondrial mass, we measured the protein levels of the proteins VDAC1 and Tom20, which are highly abundant in the outer mitochondrial membrane. We found a decrease in VDAC1 and Tom20 levels in v-ATPase-inhibited cells (*Figure 3—figure supplement 1E*). Next, we tested if the mitochondrial network was affected by v-ATPase inhibition, and observed a robust phenotype of mitochondrial fragmentation and swelling (*Figure 3E*).

Because autophagy is impaired in fibroblasts treated with v-ATPase inhibitors but not in deferoxamine-treated cells (*Figure 2—figure supplement 1B*), and the effect on mitochondrial mass is similar in vATPase-inhibited or deferoxamine-treated cells, increased mitophagy cannot explain the decrease in mitochondrial mass. Therefore, we tested if the transcriptional program of mitochondrial biogenesis was affected. We measured the transcript levels of several mitochondrial respiratory chain subunits (all encoded in the nuclear DNA), and found that they all presented a robust down-regulation when treated with bafilomycin (*Figure 3F* and *Figure 3—figure supplement 1G*), or with saliphe (*Figure 3—figure supplement 1H*). A similar effect was observed when the fibroblasts were treated with deferoxamine, as previously shown by Pagliarini and colleagues (*Rensvold et al., 2013*). Notably, supplementation of the cell medium with Fe-citrate returned the transcript levels of the mitochondrial respiratory chain subunits to control levels in bafilomycin (*Figure 3G*) or saliphe treatment (*Figure 3—figure supplement 1I*). The protein levels of several mitochondrial subunits also show a recovery to control levels when Fe-citrate is given together with bafilomycin (*Figure 3—figure supplement 1J*).

Finally, we also assessed the efficiency of electron transfer along the mitochondrial respiratory chain by estimating the levels of superoxide ($O_2^-\cdot$, the product of a single-electron reduction of $O_2$), which is a byproduct of inefficient electron transfer along the chain. Using the mitochondria-targeted superoxide-sensitive dye MitoSox, we observed a sharp increase in MitoSox levels in bafilomycin-treated fibroblasts, almost completely recovered in the fibroblasts that also received Fe-citrate (*Figure 3H*). We probed the levels of pro-oxidant molecules (superoxide, hydroxyl radical, among

others) using the probe dichlorofluoresceine (DCF), which reacts with many reactive oxygen species (ROS). We observed a sharp increase in DCF intensity in bafilomycin-treated cells, which was normalized by co-treatment with Fe-citrate (*Figure 3—figure supplement 1K*). A similar pattern is observed with the dye BODIPY C11, which reports the levels of oxidized membrane phospholipids (*Figure 3—figure supplement 1L*). The increased levels of mitochondrial superoxide, ROS and lipid peroxidation were recapitulated in saliphe-treated fibroblasts and rescued following co-treatment with Fe-citrate (data not shown). Overall, these results show that mitochondrial biogenesis and function require iron, made available to the cytoplasm by the endolysosomal system or via direct transport through the plasma membrane (Fe-citrate supplementation). In the absence of iron, mitochondrial function is impaired and generates more superoxide, promoting an oxidative shift in the cellular redox environment.

We then set to test whether the cells with genetic perturbation of lysosomal acidification also presented mitochondrial defects. Using the *Atp6v1h*-silenced fibroblasts, we observed a robust decrease in their mitochondrial respiratory chain activity (*Figure 3—figure supplement 2A*), decreased protein levels of several OXPHOS subunits (*Figure 3—figure supplement 2B*) and decreased mitochondrial biogenesis (*Figure 3—figure supplement 2C*). Taken together, genetic and pharmacologic impairment of lysosomal acidification yields similar effects on iron homeostasis and mitochondrial function.

Because retention of iron, due to impaired lysosomal acidification, triggers iron deficiency and mitochondrial dysfunction, we sought to test if the removal of lysosomal iron exporters, independently of acidification, could recapitulate the phenotype. Iron can be released from the lysosomes by DMT1 (SLC11A2) or MCOLN1 (*Dong et al., 2008*; *Touret et al., 2003*). Therefore, we used fibroblasts obtained from *Mcoln1*-KO mice (and WT littermates), and silenced *Slc11a2* using siRNA. We confirmed the efficiency of *Slc11a2* silencing by qPCR (*Figure 3—figure supplement 2D*). The individual loss of *Mcoln1* or *Slc11a2*, as well as the simultaneous loss of both, triggered an increase in the transcript levels of *Tfrc* (*Figure 3—figure supplement 2E*) and a decrease in the protein levels of FTL1 (*Figure 3—figure supplement 2F*). Mitochondrial respiration was robustly inhibited in the absence of *Mcoln1* and/or *Slc11a2* (*Figure 3—figure supplement 2F*), which was underscored by the decrease in the protein levels (*Figure 3—figure supplement 2G*) and transcript levels of OXPHOS subunits (*Figure 3—figure supplement 2H*). Altogether, these results show that iron retention in the lysosome results in functional iron deficiency and decreased mitochondrial biogenesis and function.

## Lysosomal Fe$^{2+}$ efflux is required for cell proliferation in a mitochondrial respiratory chain-dependent manner

Given the importance of iron to many cellular processes, we tested whether iron-deficiency induced by v-ATPase inhibition could affect cell proliferation. We observed that fibroblasts treated with vehicle control proliferate, while both bafilomycin (*Figure 4A*) and saliphe treatments (*Figure 4B*) resulted not only in halting of cell proliferation but also in cell death (the number of cells decreases during the treatment). The treatment with saliphe has slightly lower impact on cell death than with bafilomycin (*Figure 4B*). Remarkably, the supplementation of the medium with Fe-citrate during v-ATPase inhibition restores cellular proliferation (*Figure 4C*). Similar results were observed in fibroblasts treated with the iron chelator deferoxamine (*Figure 4—figure supplement 1A–B*).

Because iron supplementation in the presence of v-ATPase inhibitors also improved mitochondrial function, we tested if the mitochondrial respiratory chain activity was necessary for the recovery in cell viability and proliferation. The addition of the mitochondrial respiratory chain complex III inhibitor antimycin A to bafilomycin-treated iron-supplemented fibroblasts ablated the recovery in cell viability conferred by the iron supplementation (*Figure 4D*). A similar result was observed in saliphe-treated (*Figure 4E*) and in deferoxamine-treated fibroblasts (*Figure 4—figure supplement 1C*).

Next, we sought to characterize the cell death phenotype in response to v-ATPase inhibition. We first looked for apoptotic markers such as cleaved (active) caspase three and cleaved PARP (caspase three substrate). We could not detect either cleaved caspase-3 or cleaved PARP in bafilomycin- (*Figure 4F*) or saliphe-treated fibroblasts (*Figure 4—figure supplement 1D*), despite they were readily detectable in cells treated with the apoptosis inducer staurosporine. Accordingly, pan-caspase inhibitor ZVAD had no attenuating effect on cell death in bafilomycin- or saliphe-treated fibroblasts (*Figure 4G*), further supporting that the cell death in these conditions is not apoptotic. We

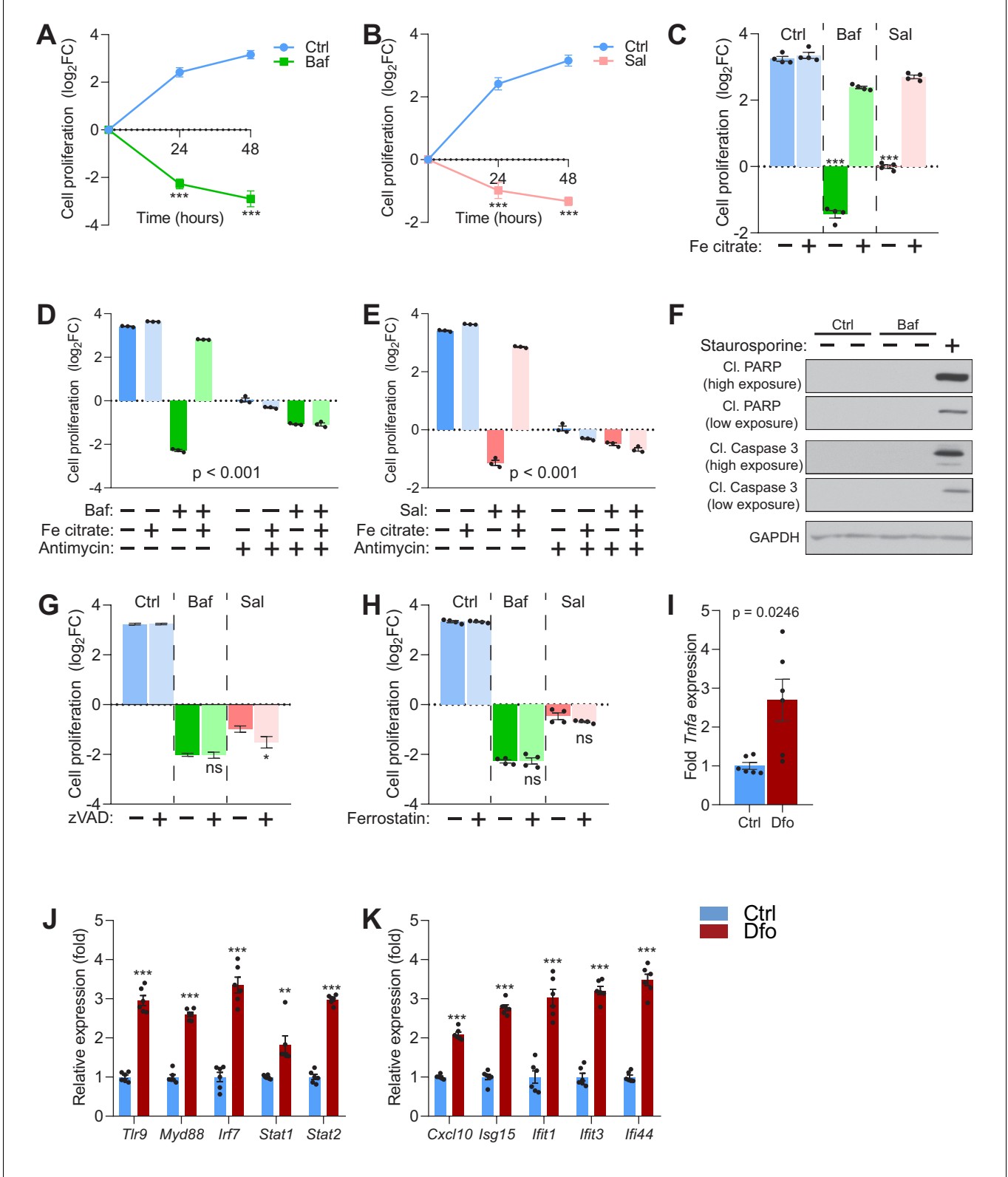

**Figure 4.** Impaired lysosomal iron efflux results in non-apoptotic cell death. (**A–B**) 500 nM Baf-treated (**A**) and 500 nM Sal-treated (**B**) fibroblasts display about 4- to 8-fold increase in cell death with time of treatment, relative to untreated fibroblasts, which show progressive cell proliferation. Results are presented as mean ± SEM, n = 4 experimental replicates with each experimental replicate being the average of technical triplicates. ***p<0.001, unpaired two-tailed t-test with Welch's correction (**C**) 150 μM iron citrate supplementation rescues cell death in fibroblasts upon treatment with 500 nM

*Figure 4 continued on next page*

Figure 4 continued

Baf or Sal. Bars and error bars represent mean ± SEM of log2 fold change in cell number of experimental conditions. Black dots indicate individual experimental measurements. \*\*\*p<0.001, Welch's one-way ANOVA with Dunnett's correction for multiple comparisons. All conditions compared to fibroblasts treated with either Baf or Sal alone. (D–E) 20 µM Antimycin treatment for 48 hr abolishes the rescue of cell death following iron supplementation in 500 nM Baf-treated (C) or 500 nM Sal-treated (D) fibroblasts. Bar graphs represent mean ± SEM of three independent experimental measures (black dots). \*\*\*p<0.001, Two-way ANOVA with Sidak correction for multiple comparisons. (F) Whole cell immunoblot of cleaved PARP and cleaved caspase-3 levels shows that Baf-induced cell death is non-apoptotic (n = 3). 1 µM Staurosporine treatment for 4 hr is used as positive control for caspase-3 dependent apoptotic cell death. GAPDH is used as loading control. (G–H) Inhibitors of apoptosis (G) and ferroptosis (H) show no observed effects on 500 nM Baf or Sal-induced cell death in fibroblasts following 48 hr of treatment. Results show mean ± SEM, n = 4. \*p<0.5; ns p>0.05, Welch's one-way ANOVA with Dunnett's correction for multiple comparisons to treatment with Baf or Sal alone. 20 uM zVAD or 5 µM Ferrostatin were used. (I–K) 300 µM Dfo treatment in neurons results in more than 2-fold increase in the expression of *Tnfa* (I), increased expression of regulators of interferon gene expression (J) and increased transcript levels of interferon-stimulated genes (K). Results show mean ± SEM of six independent experiments. \*\*p<0.01; \*\*\*p<0.001, unpaired two-tailed t-test with Welch's correction.

The online version of this article includes the following figure supplement(s) for figure 4:

**Figure supplement 1.** Iron deficiency-induced cell death is non-apoptotic.

then tested if the cell death induced by v-ATPase inhibition was sensitive to necrostatin (inhibitor of necrosis) or to ferrostatin (inhibitor of ferroptosis). Necrostatin-1s, which can decrease necroptotic cell death by inhibiting RIP1 kinase, had no effect on the cell death caused by v-ATPase inhibition (*Figure 4—figure supplement 1E*). Ferrostatin, an inhibitor of ferroptosis, also had no effect on v-ATPase inhibition-induced cell death (*Figure 4H*), despite it effectively inhibited cell death triggered by the ferroptosis inducer erastin (*Figure 4—figure supplement 1F*). Therefore, v-ATPase inhibition triggers caspase-independent, RIP1K-independent, ferrostatin-unrelated cell death. These results are consistent with the previously reported caspase-independent bafilomycin-induced cell death (*Yan et al., 2016*). In agreement with this finding, the analysis of the three transcriptome datasets of bafilomycin-treated cells that were used in *Figure 1* ranks necrosis (and not apoptosis) as the top cell death mechanism (*Supplementary files 1–3* tables S1-S3).

## Iron deficiency triggers cell-autonomous inflammatory gene expression

The occurrence of non-apoptotic cell death is often associated with triggering of sterile inflammatory responses (*Rock and Kono, 2008*; *Weinlich et al., 2017*). To test if iron deficiency is sufficient to trigger immune responses, we treated mouse primary cortical neurons with deferoxamine, and monitored the expression of pro-inflammatory cytokines and interferon-stimulated genes, which are typically increased during inflammatory responses. We first observed the expression of tumor necrosis α (*Tnfa*) and found a robust increase in its transcript levels in deferoxamine-treated primary neurons (*Figure 4I*). We then measured the transcript levels of innate immune regulators such as *Tlr9*, *Myd88*, *Irf7*, *Stat1* and *Stat2*, which were all robustly up-regulated in deferoxamine-treated cortical neurons (*Figure 4J*). Accordingly, the expression of other downstream interferon-stimulated genes such as *Cxcl10*, *Isg15*, *Ifit1*, *Ifit3* and *Ifi44* was also strongly induced in deferoxamine-treated neurons (*Figure 4K*). We have not tested the effect of bafilomycin or saliphe treatment in the primary neurons because these v-ATPase inhibitors would not just inhibit lysosomal function but also synaptic vesicle recycling (*Farsi et al., 2018*), and therefore trigger a number of events that are unrelated to iron homeostasis alone. These results underscore that unavailability of cytoplasmic iron is sufficient to trigger inflammatory signaling and cell death.

## Iron deficiency is prevalent in the brain of an in vivo model of impaired lysosomal acidification

Given the pro-inflammatory nature of Fe-deficiency caused by lysosomal v-ATPase inhibition, we sought to test the system in vivo, using a mouse model lacking *Gaa* (*Gaa*-KO), as described above. Absence of *Gaa* impairs the acidification of a large proportion of lysosomes, and fibroblasts obtained from these mice show iron deficiency and HIF-1α accumulation (see *Figure 1F–I*). The *Gaa*-KO mice show a predominantly muscular phenotype, albeit with a later disease onset when compared to human patients with *Gaa* loss-of-function mutations (*Raben et al., 1998*). Nevertheless, the mice present severe motor symptoms with onset around 14 months, and so we assessed them at earlier time points (6 and 12 months) to avoid this confounding factor. We focused on the brain of

the *Gaa*-KO mice, as loss of lysosomal acidification has been observed in association with neurodegenerative diseases (*Lie and Nixon, 2019*). Furthermore, it has been shown that the muscular phenotype of the *Gaa*-KO mice can be rescued by re-expression of *Gaa* in the motor neurons (*Lee et al., 2018*; *Todd et al., 2015*; *Turner et al., 2016*), illustrating that lysosomal function is relevant in neuronal cell populations. First, we tested if the cortex of the *Gaa*-KO mice showed signs of iron deficiency. We observed a robust increase in the transcript levels of *Tfrc* in the cortex of 6- and 12-month-old mice (*Figure 5A*). Ferritin light chain protein levels were down-regulated both at 6- and 12 months (*Figure 5B*). Together, these results document an iron deficiency response in *Gaa*-KO brain. To further probe the functional consequences of iron-deficiency in the *Gaa*-KO brain, we also tested if HIF-1α was accumulating, by western blot, and found an increase in its levels in *Gaa*-KO cortex both at 6- and 12 months (*Figure 5C*). Furthermore, mitochondrial iron availability was also affected, as the activity of complex IV, which is dependent on Fe, was decreased in *Gaa*-KO cortex (*Figure 5D*), while the activity of complex V (loading control) was similar between WT and KO mice (*Figure 5D*). The decrease in complex IV activity (cytochrome c oxidase, COX) occurs despite the protein levels of the subunit COX1 were similar between WT and KO both in whole cell extracts (*Figure 5—figure supplement 1A*) and in mitochondrial extracts from mouse cortices (*Figure 5—figure supplement 1B*).

We then performed in vivo magnetic resonance spectroscopy (MRS) and imaging in the mice at 6 and 12 months of age (*Figure 5E* and *Figure 5—figure supplement 1D*), to assess if the perturbations in iron homeostasis were widespread or in discrete brain regions. For this, we analyzed the thalamus, striatum and cerebral cortex, and in all three regions we found a decrease in the $T_2$ relaxation time of water protons (*Table 1*), which is highly correlated with reduced labile iron pool levels (*Vymazal et al., 1993*). Interestingly, we also observed by MRS, reduced levels of choline-containing compounds, which may be due slow turnover of phosphatidylcholine, further suggesting impaired myelination (*Figure 5E–F* and *Supplementary file 4*).

We therefore tested whether myelination (which is also a Fe-dependent process) was affected in the *Gaa*-KO cortex. We observed that the abundance of myelination-related proteins such as proteolipid protein (PLP) and myelin basic protein (MBP) was decreased in *Gaa*-KO cortical homogenates (*Figure 5—figure supplement 1C*). Altogether, these results show that the *Gaa*-KO brain is iron-deficient, with consequences for HIF signaling, mitochondrial function and myelination.

## *Gaa*-KO brain shows prevalent inflammation at early presymptomatic stages

Having shown that the *Gaa*-KO brain presents functional iron deficiency akin to what was observed in cultured *Gaa*-KO fibroblasts, as well as in fibroblasts treated with v-ATPase inhibitors or fibroblasts with genetic silencing of vATPase subunits, we sought to determine if we could detect inflammatory signatures in the *Gaa*-KO mouse cortex. In order to have an unbiased approach, we performed RNA sequencing of WT (n = 5) and *Gaa*-KO (n = 5) 12 month-old cortices. We identified 1779 differentially expressed genes (adjusted p value < 0.05; fold change >2.0), of which 996 were up-regulated and 783 were down-regulated in *Gaa*-KO cortices. The differential gene list fully segregates WT and KO samples in hierarchical clustering (*Figure 6A*). The most significantly changed transcripts were *Gaa* (as expected) and several proteins related to immune responses, particularly complement activation, macrophage infiltration and genes induced by interferon signaling (*Figure 6B*). Accordingly, several pathways related to inflammation were found enriched in the *Gaa*-KO dataset (*Figure 6C*). We then determined which upstream regulators (specifically, transcription factors) were affected in the *Gaa*-KO cortex, and found that the interferon regulatory factors *Irf7* and *Irf3* were predicted as the two most active transcription factors (*Figure 6D*). Thus, we focused on the target genes of *Irf7*, and using the *Gaa*-KO cortex transcriptome data we observed that 17 out of 20 *Irf7* targets in the *Gaa*-KO brain were robustly up-regulated (*Figure 6E*). Altogether, these results show in an unbiased manner that the 12 month old cortex of the *Gaa*-KO mouse presents a robust inflammatory signature involving *Irf7* signaling.

Next, we tested if the inflammatory signature in the *Gaa*-KO cortex can be detected at earlier ages. We measured by qPCR the transcript levels of several transcripts regulated by *Irf7*, most of which were up-regulated in the *Gaa*-KO brain at 2, 6 and 12 months of age (*Figure 6F*). The exception was *Stat1*, which was up-regulated only at 12 months.

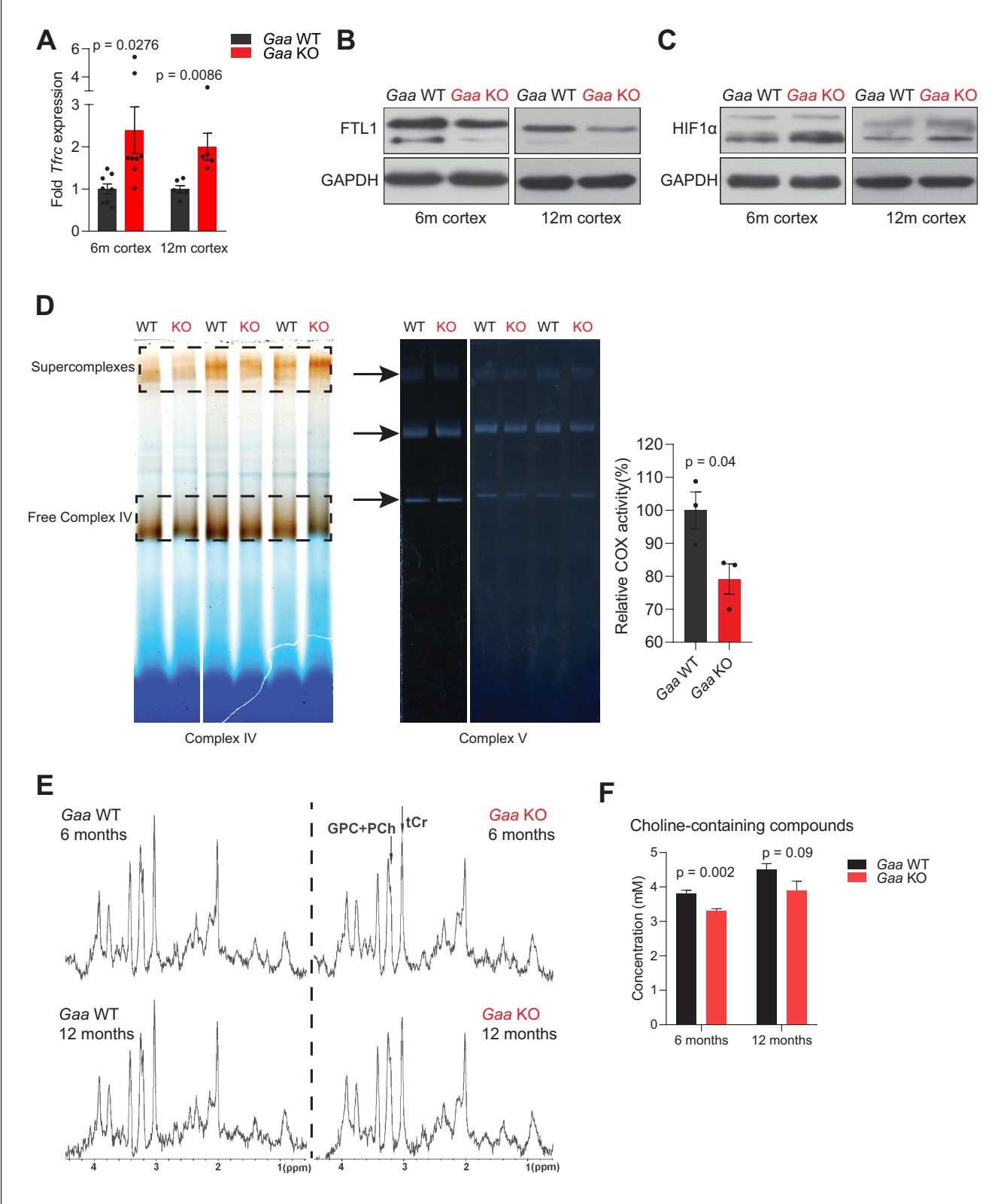

**Figure 5.** Functional iron deficiency in a mouse model of impaired lysosomal acidification (*Gaa*<sup>-/-</sup> mouse). (**A**) *Tfrc* mRNA levels in the cortex of 6 and 12 month old *Gaa*<sup>-/-</sup> mice and their wild type littermate controls. Black dots represent sample size (n) in each group. p values are unpaired two-tailed t-tests with Welch's correction. Bars are depicted as mean ± SEM. (**B**) Western blot of FTL1 and GAPDH as loading control from cortical tissue lysates of 6 and 12 month old *Gaa*<sup>+/+</sup> and *Gaa*<sup>-/-</sup> mice (n = 4 mice per group). (**C**) HIF-1α immunoblots with GAPDH as loading control from cortical tissue lysates

*Figure 5 continued on next page*

*Figure 5 continued*

of 6 and 12 month old $Gaa^{+/+}$ and $Gaa^{-/-}$ mice (n = 4 mice per group). (**D**) Activity staining of native respiratory chain complex IV of mitochondria purified from cortices of 6 month old mice (n = 3; depicted as black dots). Complex V activity staining is used as loading control. The difference between activities (mean ± SEM) of groups is determined by the unpaired two-tailed t-test with Welch's correction. (**E**) MRS reveals significant changes in metabolite concentrations in $Gaa^{-/-}$ mice in vivo. MRS (STEAM, TR/TE/TM = 6000/10/10 ms, 256 averages, (2.0 mm)$^3$ volume-of-interest centered on the striatum) of 6 month old (upper row) and 12 month old (lower row) wild type (right column) and $Gaa^{-/-}$ mice (left column) in vivo, as summarized in *Table 1*, processed with a 1 Hz line broadening. GPC+PCh = Choline containing compounds, tCr = total creatine, ↑=significant signal intensity increases, ↓=significant signal intensity decreases. *p<0.05; **p<0.01, Mann-Whitney's U-test. (**F**) Graphs depicted as mean ± SEM of n = 8 mice per group (6 months) or five mice per group (12 months) show decreased concentration of total choline-containing compounds in $Gaa^{-/-}$ mice in vivo. p values indicate the Mann-Whitney's U-test.

The online version of this article includes the following figure supplement(s) for figure 5:

**Figure supplement 1.** Steady state levels of COX1, PLP and MBP proteins, and in vivo MRI of mouse brain.

Glial activation is a typical marker of brain inflammation, and can be assessed by the levels of the microglia- and astrocyte-enriched protein GFAP (which is highly up-regulated at transcript level in the RNA sequencing dataset, see *Figure 6B*). Therefore, we searched for markers of glial activation in the *Gaa*-KO cortex. We observed an increase in GFAP protein levels in GAA-KO cortex at 6 and 12 months (*Figure 6G*), by western blot. Staining of tissue sections against GFAP protein in the 6 month GAA-KO cortex yielded similar results (*Figure 6H*).

Altogether, these results show that the brain of GAA-KO exhibits increased inflammatory and interferon signatures and enrichment of innate immune cells detectable already at 2 months of age.

## Iron deficiency perturbs mtDNA homeostasis in vivo

Mitochondrial malfunction has been shown to be a contributor to inflammatory reactions (*West and Shadel, 2017*). In particular, imbalance in mtDNA homeostasis is a robust trigger of type I interferon responses (*West et al., 2015*). Therefore, we sought to test if iron deficiency, which we showed to be sufficient to trigger inflammation in cultured neurons (see *Figure 4I–K*), was associated with mtDNA perturbation in vivo. First, we measured mtDNA copy number in the *Gaa*-KO cortex. We observed a reduction in mtDNA of about 20% in newborn mice, 25% at 2 months, and a robust decrease of ~50–60% in 6- and 12 month old *Gaa*-KO cortices (*Figure 7A*). Notably, the protein levels of TFAM, a key protein for mtDNA replication, transcription and maintenance, were reduced by ~50% in the 6 month-old *Gaa*-KO cortex (*Figure 7B*). This decrease is akin to the loss of TFAM and mtDNA observed in *Tfam* heterozygous knockout mice, in which mtDNA instability was shown

**Table 1.** Relaxation times of water protons and Magnetization Transfer ratios determined by MRI (related to *Figure 5*).

| | 6 months | | 12 months | |
|---|---|---|---|---|
| **Background** | **Wild type** | **Gaa(-/-)** | **Wild type** | **Gaa(-/-)** |
| *n* | *n* = 8 | *n* = 8 | *n* = 5 | *n* = 5 |
| *T$_1$* (s) | | | | |
| Cerebral Cortex | 1.67 ± 0.07 | 1.67 ± 0.07 | 1.65 ± 0.05 | 1.62 ± 0.09 |
| Striatum | 1.62 ± 0.05 | 1.64 ± 0.07 | 1.62 ± 0.03 | 1.58 ± 0.06 |
| Thalamus | 1.49 ± 0.03 | 1.52 ± 0.09 | 1.50 ± 0.04 | 1.46 ± 0.03 |
| *T$_2$* (ms) | | | | |
| Cerebral Cortex | 40.3 ± 0.5 | 38.3 ± 0.8*** | 39.8 ± 0.7 | 37.9 ± 0.5* |
| Striatum | 39.7 ± 1.0 | 37.3 ± 0.8** | 38.5 ± 0.6 | 36.7 ± 0.7* |
| Thalamus | 38.4 ± 1.0 | 35.9 ± 1.1** | 37.2 ± 0.3 | 36.2 ± 0.8 |
| *Magnetization Transfer Ratio (%)* | | | | |
| Cerebral Cortex | 7.5 ± 1.7 | 7.0 ± 1.7 | 7.8 ± 0.9 | 8.4 ± 0.6 |
| Striatum | 9.9 ± 0.7 | 9.0 ± 0.8* | 9.8 ± 0.6 | 10.0 ± 0.9 |
| Thalamus | 12.5 ± 1.6 | 11.9 ± 1.9 | 12.9 ± 2.0 | 13.0 ± 0.9 |

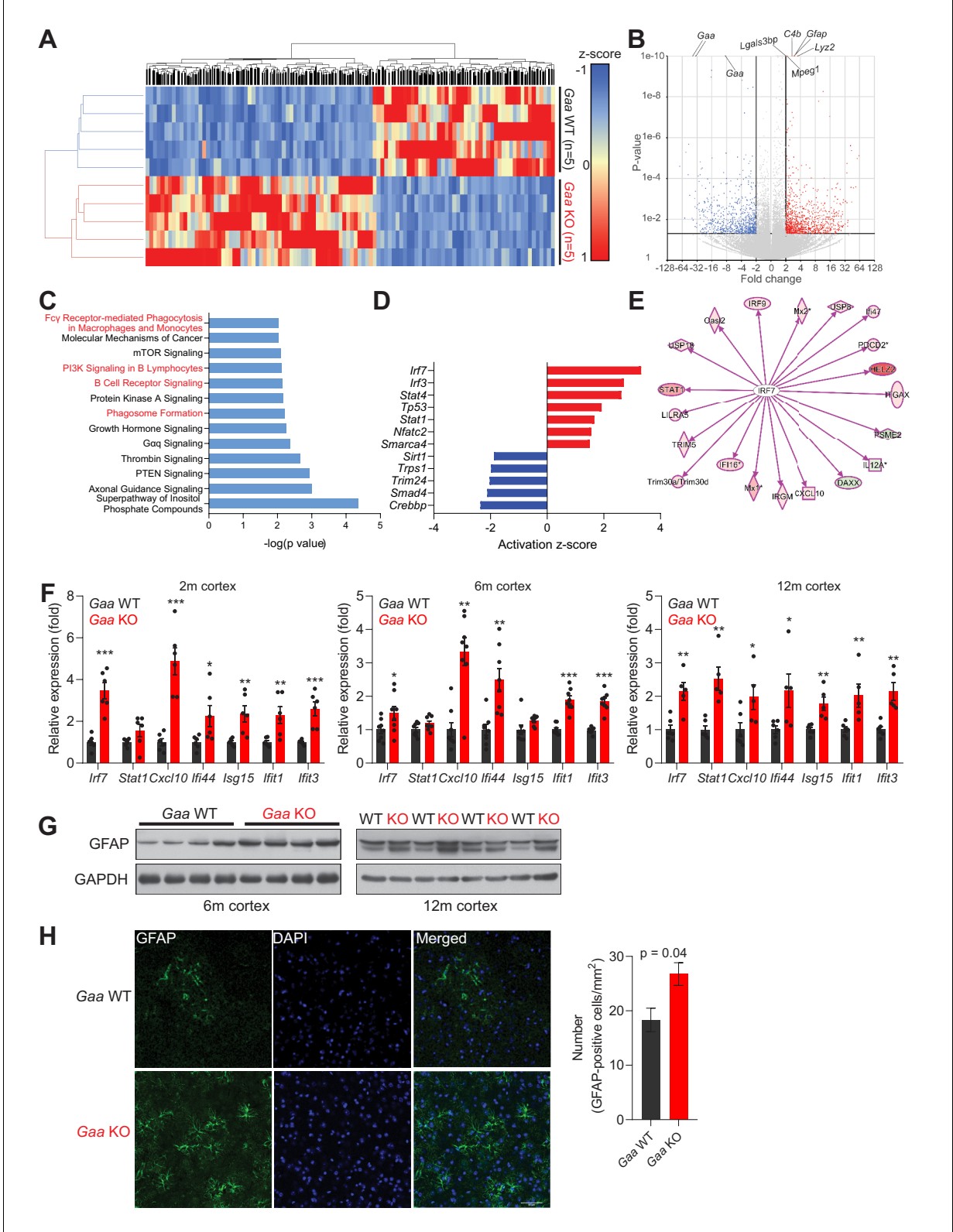

**Figure 6.** Iron deficiency is linked to induction of innate immunity in the CNS of *Gaa*[-/-] mice. (**A**) Hierarchical clustering of the brain samples of *Gaa*[-/-] and WT mice at 12 months of age. (**B**) Volcano plot of the transcripts detected in the RNAseq analysis, highlighting those that were considered as differentially expressed genes (adjusted p value < 0.05, and fold-change >2). Some of the transcripts with the lowest p value between WT and KO are highlighted in the plot. (**C**) Results of the pathway analysis showing the pathways most enriched in the differentially expressed gene list of the *Gaa*-KO/

*Figure 6 continued on next page*

*Figure 6 continued*

WT cortex dataset. The pathways related to inflammation are highlighted in red. (D) Transcription factors predicted to be significantly active or repressed in the cortex of 12 month old *Gaa⁻/⁻* mice compared to WT. The z-score is represented (positive z-score, predicted activation; negative z-score, predicted repression). (E) Transcript levels of Irf7 targets recognized in the *Gaa⁻/⁻*dataset by Ingenuity Pathway Analysis. The targets are color-coded (red indicates higher expression in *Gaa⁻/⁻*mice, green represents lower expression), showing that the vast majority is up-regulated, in agreement with the predicted activation of *Irf7*. (F) Transcript levels of interferon-stimulated genes and their regulators in the cortices of 2 month, 6 month and 12 month old *Gaa⁻/⁻* mice and their wild type littermate controls. Number of mice per group is displayed as black dots in bars, which represent mean ± SEM. *p<0.05; **p<0.01; ***p<0.001, unpaired two-tailed t-test with Welch's correction. (G) Whole cortical tissue immunoblots of increased GFAP levels and GAPDH as loading control in 6- and 12 month old wild type and *Gaa⁻/⁻* mice (n = 4 mice per group). (H) Representative images of GFAP (green) in the cortex of 6 month old *Gaa⁻/⁻* mice and their wild type littermate controls. Scale bar, 20 µm. Quantifications on the right show the increased number of GFAP-positive cells in the cortices of *Gaa⁻/⁻* mice (n = 3 mice per group) depicted as bars denoting mean ± SEM. p value is estimated from an unpaired two-tailed t-test with Welch's correction.

to trigger interferon signaling (*West et al., 2015*). The decrease in TFAM protein levels cannot be explained only by decreased transcription, since *Tfam* transcript levels are only 20% down in the *Gaa*-KO brain (*Figure 7—figure supplement 1A*).

We then turned to clonal cells to test if the loss of mtDNA observed in vivo could be triggered in cells by iron deficiency. We first measured mtDNA copy number in WT fibroblasts treated with the lysosomal iron chelator deferoxamine, and observed a reduction of ~40% in mtDNA (*Figure 7—figure supplement 1B*). A similar result was obtained by deferoxamine treatment of primary cortical neurons (*Figure 7—figure supplement 1C*).

To test the effect of iron deficiency on mtDNA in the absence of lysosomal perturbations, we used fibroblasts obtained from mice lacking the iron regulatory proteins IRP1 and IRP2, in which the expression of heavy and light ferritin chains is robustly up-regulated, and for that reason are functionally iron deficient (all available iron is stored in the ferritin oligomers, and thus biologically inactive) (*Meyron-Holtz et al., 2004*). The mtDNA copy number in the *Irp1⁻/⁻* and *Irp2⁻/⁻*KO fibroblasts was ~40% reduced compared to WT (*Figure 7—figure supplement 1D*). We then tested whether impaired lysosomal acidification caused by genetic mutations also resulted in decreased mtDNA copy number. To this end, we used fibroblasts obtained from *Gaa*-KO and respective WT littermates, and observed a loss of ~40% mtDNA in *Gaa*-KO (*Figure 7—figure supplement 1E*). Notably, the protein levels of TFAM were also decreased in *Gaa*-KO fibroblasts (*Figure 7—figure supplement 1F*).

mtDNA synthesis relies on dNTPs imported from the cytoplasmic dNTP pool (*Copeland, 2012*), which is high in proliferating cells but decreases sharply in post-mitotic cells and tissues. Ribonucleotide reductase is a key enzyme for the reduction of NTPs to dNTPs, and its active site is dependent on iron (*Puig et al., 2017*). Thus, the iron deficiency observed in *Gaa*-KO cells or Deferoxamine-treated cells might result in a decrease in functionally active ribonucleotide reductase. Therefore, we supplemented the cell medium with Fe-citrate, and observed that the mtDNA copy number increased in iron-supplemented *Gaa*-KO fibroblasts to control values (*Figure 7C*).

Having shown that mtDNA levels can be rescued in iron-deficient cells by Fe-citrate supplementation in the medium, we sought to test if the same principle was valid in vivo. Therefore, we tested if increasing the concentration of iron in the diet would avoid loss of mtDNA in vivo, as well as ameliorate the downstream consequences of iron-deficiency in *Gaa*-KO mice. The change in iron levels in the diet is most efficient when it occurs at weaning. Thus, upon separating the litters from the mothers (at post-natal day 21), we gave the weaned mice an iron-enriched diet (500 mg Fe/Kg) or a control diet (179 mg Fe/Kg). We followed the mice until they were two months of age, and measured mtDNA copy number in the cortex. While the GAA-KO mice treated with the control Na-citrate diet showed a ~ 20% decrease in mtDNA in cortex relative to the WT littermates, the *Gaa*-KO cohort fed iron-enriched chow had mtDNA copy number similar to the WT mice (*Figure 7D*). This result shows that the mitochondrial phenotype of young *Gaa*-KO mice can be rescued by increasing the levels of dietary iron.

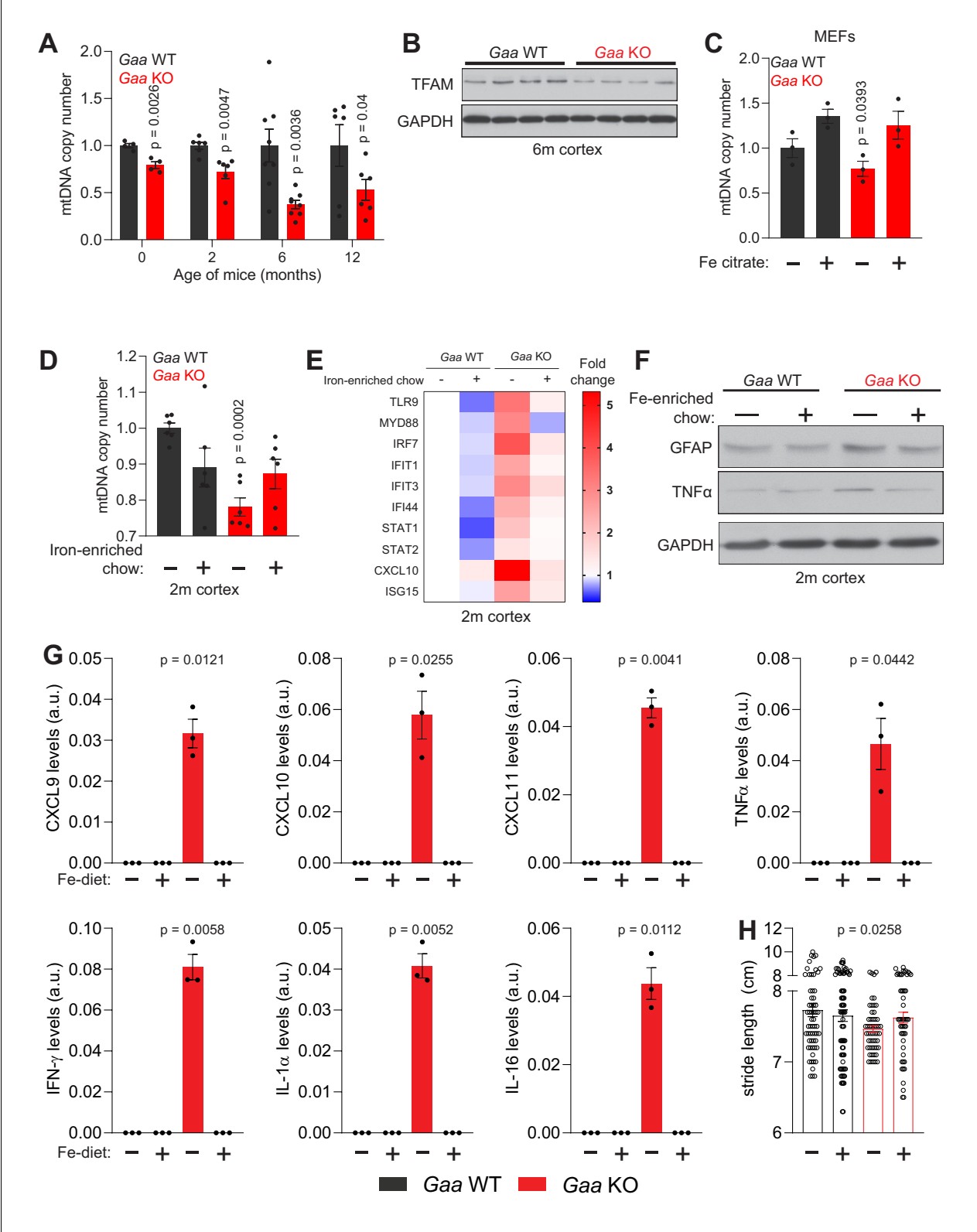

**Figure 7.** Iron deficiency impairs mtDNA homeostasis in *Gaa*[-/-] mice. (A) Age-dependent decline in relative mtDNA copy number levels in the cortex of *Gaa*[-/-] mice from 0 to 12 months of age. Number of mice per group is depicted as dots in bars which denote mean ± SEM. *p<0.05; **p<0.01; ***p<0.001, unpaired two-tailed t-test with Welch's correction. (B) Western blot showing reduced TFAM and GAPDH as loading control in the cortex of *Gaa*[-/-] mice and their wild type littermate controls (n = 8 mice per group). (C–D) Iron supplementation rescues mtDNA copy number defects in *Gaa*[-/-]

*Figure 7 continued*

fibroblasts (C) and in the cortex of 2 month old *Gaa*^-/- mice (D). Results are summarized as mean ± SEM of n = 3 independent measures in *Gaa*^-/-
fibroblasts and n = 6 mice per group indicated as black dots for 2 month old *Gaa*^-/- mice. *p<0.05; ***p<0.001, Welch's one-way ANOVA test with
Dunnett's correction for multiple comparisons. All conditions compared to the untreated *Gaa*^-/- fibroblasts (C) or *Gaa*^-/- mice fed standard chow (D). (E)
Iron supplementation as iron-enriched chow dampens the increased expression of innate immune response genes in *Gaa*^-/- mice at 2 months of age.
This result is depicted as a Heatmap for n = 6 mice per group, in which blue denotes decreased expression of interferon-stimulated genes relative to
white (no change or baseline) and red denotes increased expression. Note the dampening of the hot red in *Gaa*^-/- on standard chow relative to the
cohort of *Gaa*^-/- mice on iron-enriched chow. (F) Immunoblot of GFAP and TNFα in the cortices of *Gaa*^-/- mice and their wild type littermate controls
fed standard or iron-enriched chow for 2 months (n = 3). Note the reduction in GFAP and TNFα protein amounts following iron supplementation in
*Gaa*^-/- mice. GAPDH is used as loading control (G) Iron-enriched diet restores increased levels of cytokine and chemokines to control levels in the
cortices of 2 month old *Gaa*^-/- mice. Cytokines are increased in *Gaa*^-/- mice fed standard chow, and are barely detectable in *Gaa*^-/- mice fed iron-
enriched diet or their wild type littermate controls. Levels of cytokines and chemokines are shown as mean ± SEM (n = 3 mice per group, black dots)
and p values represent Brown-Forsythe's one-way ANOVA tests estimated as differences between *Gaa*^-/- mice on standard chow and all other
experimental groups. (H) Mouse gait analysis shows rescue of stride length following 2 months of iron supplementation in *Gaa*^-/- mice. Data is
presented as mean ± SEM. Each experimental data point (hollow circles) represents stride length measured on all four limbs of each animal. 13–19
animals per group were used. p value is determined by the Welch's one-way ANOVA followed by Dunnett's correction for multiple comparisons of all
groups to *Gaa*^-/- mice fed standard diet.

The online version of this article includes the following figure supplement(s) for figure 7:

**Figure supplement 1.** Iron deficiency causes impaired mtDNA homeostasis and induces inflammation in vivo.

## Dietary iron supplementation corrects inflammation in the *Gaa*-KO brain

The rescue of mtDNA levels in *Gaa*-KO brain by using a Fe-enriched diet encouraged us to test if other parameters responding to iron deficiency would also be corrected, in particular inflammation. So, we analyzed the transcript levels of interferon-stimulated genes and those of their regulators in the cortex of 2 month old mice in Fe-enriched or control diet, and observed that in the iron-supplemented *Gaa*-KO mice the inflammatory response was ablated (*Figure 7E*). Furthermore, we observed that the protein levels of GFAP and of the inflammatory mediator TNFα were also decreased by the iron-supplementation in the *Gaa*-KO cortex (*Figure 7F*).

Next, we sought to characterize the molecules driving the inflammatory response. To achieve that, we employed a cytokine array, and used *Gaa*-KO and WT 2 month old cortical extracts, from mice that were either iron-supplemented or in the control diet. While we barely could observe TNFα, interferon gamma (IFNγ) as well as the cytokines CXCL9, CXCL10, CXCL11, IL-1α, IL-16 in wild type mice, these proteins were significantly increased in the cortices of *Gaa*-KO fed standard chow, but were normalized to the barely detectable WT levels in the iron-supplemented mice (*Figure 7G*). Other inflammatory molecules, including TREM-1, M-CSF, TIMP-1, C5, ICAM-1 and CXCL12 showed a similar profile (*Figure 7—figure supplement 1G–H*).

These results show that the pro-inflammatory phenotype observed in the brain of the *Gaa*-KO mice is due to the impaired iron homeostasis and can be corrected by increasing iron concentration in the diet.

To test whether the dietary iron supplementation had any effects on the behavior of the mice, we made a broad analysis of gait and motor parameters of the mice using the DigiGait platform (*Rostosky and Milosevic, 2018*). We observed that several parameters were affected in the *Gaa*-KO already at 2 months of age, particularly the stride length, which was decreased in the *Gaa*-KO mice (*Figure 7H*). This is an indication of gait disorder (*Fernagut et al., 2002*), suggesting that the *Gaa*-KO mice may have motor impairments. Notably, the *Gaa*-KO mice supplemented with iron showed a normalization of the stride length (*Figure 7H*). Taken together, normalization of iron levels by a Fe-enriched diet restores mtDNA stability and ablates inflammation at 2 months of age in a mouse model of lysosomal malfunction causing functional iron deficiency.

## Discussion

This study shows that iron deficiency can cause non-apoptotic cell death and inflammation in cultured cells as well as inflammation in vivo. Furthermore, we establish the role of (endo)lysosomal acidification as a key step in cellular iron homeostasis. Notably, the inflammatory signaling triggered

by (endo)lysosomal malfunction-induced iron deficiency can be ablated by supplementation of iron that bypasses the endo-lysosomal pathway.

We show here that iron release from (endo)lysosomes to cytoplasm requires acidic pH. This step is key to maintaining appropriate cytoplasmic iron levels. When the (endo)lysosomes are not properly acidified, $Fe^{3+}$ obtained from ferritinophagy, mitophagy or transferrin endocytosis cannot be reduced to $Fe^{2+}$. This reduction is catalyzed by the ferroreductase STEAP3, which has an optimal acidic pH (*Lambe et al., 2009*). The inability to reduce $Fe^{3+}$ to $Fe^{2+}$ results in the accumulation of $Fe^{3+}$ in the lysosomes, since only $Fe^{2+}$ can be exported via the channels divalent metal transporter 1 (DMT1) or mucolipin-1 (MCOLN1) (*Dong et al., 2008*; *Touret et al., 2003*). In addition to the excess of iron in the lysosomes, there is lack of it in the cytoplasm. The proteins that sense cellular iron function are located in the cytoplasm, implying that cytoplasmic Fe levels determine when the cellular iron deficiency response is activated. Low Fe levels result in the conversion of aconitase in IRP1 and stabilization of IRP2 (*Rouault, 2015*). Together, these proteins promote iron mobilization and uptake, while repressing its storage and export from the cell. This program is carried out by repressing the translation of ferritin heavy and light chains (to inhibit storage), and by stimulating the expression of transferrin receptor (TFRC), in order to increase the uptake of iron. This response keeps Fe within a tight range of concentrations. We show here that this program of iron homeostasis requires functional lysosomes. When lysosomes are defective, either by treatment with v-ATPase inhibitors or by genetic mutations that impair their acidification, the iron homeostatic cycle is also impaired because iron cannot be released in sufficient amounts to the cytoplasm.

Iron is an essential microelement, and life relies on iron availability (*Rouault, 2015*). Accordingly, inhibition of lysosomal v-ATPase and the consequent iron deficiency result in cell death. It can be argued that this is due to toxicity of the inhibitors used, or to effects in other cellular organelles (the v-ATPase is also present at the Golgi and in synaptic vesicles) or processes. Nevertheless, the simple repletion of iron through an (endo)lysosome-independent pathway was sufficient to restore cell proliferation almost to control levels. Therefore, the main contributor to cell death caused by impaired (endo)lysosomal acidification seems to be iron deficiency. We determined that the cell death caused by iron deficiency is non-apoptotic, which likely contributes to the inflammation observed in vivo in the brain of a mouse model of lysosomal malfunction (*Gaa*-KO). In addition, iron deficiency was sufficient to trigger inflammatory signaling in cultured primary cortical neurons. While we cannot exclude the possibility that neuronal cell death in culture was sensed by astrocytes or glia that are present in the culture (primary neurons typically have a small number of co-cultured astrocytes and microglia), our results suggest that the inflammation trigger is cell-intrinsic.

Many cellular processes rely on iron, from DNA replication (the synthesis of dNTPs is Fe-dependent) to metabolism (mitochondrial respiratory chain relies on Fe-S clusters) (*Rouault, 2015*). Importantly, mitochondrial respiratory chain and oxidative phosphorylation subunits are encoded by two genomes, the nuclear DNA and the mitochondrial DNA (mtDNA). There is no dedicated de novo production of dNTPs for mtDNA synthesis, and mtDNA replication relies mostly on the cytoplasmic dNTP production (*Copeland, 2012*). A key step in dNTP production is catalyzed by the enzyme ribonucleotide reductase, a heterodimeric protein that has Fe in its active center (*Puig et al., 2017*). Decreased availability of dNTPs is a known cause of mtDNA instability, which in turn is an inflammatory trigger. We show here that iron deficiency perturbs mtDNA homeostasis, regardless of the cell type (fibroblasts, neurons) or the mechanism (genetic, pharmacologic) that leads to iron deficiency. We also show here that the levels of mtDNA decrease sharply in iron deficiency. This effect is akin to what is observed in the mice lacking one allele of the mitochondrial transcription factor A (TFAM), a protein needed to replicate, transcribe and stabilize mtDNA (*Ekstrand et al., 2004*; *West et al., 2015*). In the *Tfam*$^{+/-}$ mice, mtDNA instability triggers interferon-dependent signaling that culminates in potent inflammatory responses (*West et al., 2015*). We show here that iron deficiency, either caused by iron chelation, genetic ablation of *Irp1* or *Irp2*, or genetic defects impairing lysosomal acidification, is associated both with mtDNA instability and inflammation, in vitro and, importantly, in vivo. Notably, supplying iron via an endo/lysosome-independent route resolved the iron deficiency response, the loss of mtDNA and the inflammatory signaling.

Another process that depends on iron is the activity of the α-ketoglutarate-dependent superfamily of dioxygenases (*Raimundo et al., 2011*). These enzymes transfer oxygen atoms to different substrates, in a manner that is dependent on $O_2$, $Fe^{2+}$ and α-ketoglutarate. Absence of any of these factors or accumulation of succinate (a product of these reactions) results in inhibition of the

dioxygenases. The prolyl hydroxylases are part of the α-ketoglutarate-dependent superfamily, and catalyze the hydroxylation of the α-subunit of hypoxia-inducible factor (HIF-1α). After hydroxylation, HIF-1α is recognized by the ubiquitin ligase VHL, and labeled for proteasomal degradation (*Raimundo et al., 2011*). HIF-1α is therefore constitutively expressed and readily degraded under normal conditions, and thus barely detectable (*Raimundo et al., 2011*). However, in the absence of VHL or when the function of prolyl hydroxylases is impaired, HIF-1α accumulates and triggers a transcriptional program known as (pseudo-)hypoxia response. Therefore, the activation of HIF signaling is a novel consequence of impaired lysosomal acidification (*Miles et al., 2017*). It is noteworthy to point out that there may be other routes by which the lysosomal function regulates HIF-1α, particularly its degradation via chaperone-mediated autophagy (*Hubbi et al., 2013*). Notably, HIF signaling can also be activated by mitochondrial malfunction, particularly loss-of-function of the citrate cycle enzymes fumarate hydratase or succinate dehydrogenase, which results in the accumulation of fumarate and succinate, and in the consequent inhibition of the prolyl hydroxylases. Therefore, HIF-1α can be seen as a sensor of the mitochondrial-lysosomal axis. Reciprocally, it can also be noted that the constitutive silencing of the (pseudo)hypoxia response requires both functional lysosomes and functional mitochondria. This aspect illustrates a specific mechanism of cooperation between different organelles, in this case, between mitochondria and lysosomes.

We and others have recently unveiled several signaling pathways by which dysfunctional mitochondria result in lysosomal impairment, or vice-versa (*Demers-Lamarche et al., 2016*; *Fernández-Mosquera et al., 2017*; *Fernandez-Mosquera et al., 2019*; *Osellame et al., 2013*; *Yambire et al., 2019*). This mutually destructive relationship between mitochondria and lysosomes is particularly detrimental in neurodegenerative diseases. For example, many of the genes that cause inherited Parkinson's disease are associated either with mitochondria or lysosomes/autophagy, and therefore the primarily-damaged organelle is likely to promote the secondary impairment of the other.

This manuscript presents a novel role of mitochondria-lysosome crosstalk in the regulation of central nervous system inflammation in vivo.

# Materials and methods

## Key resources table

| Reagent type (species) or resource | Designation | Source or reference | Identifiers | Additional information |
|---|---|---|---|---|
| Antibody | Anti-ATP5B | Self-made | PRAB4826 | WB (1:1000) |
| Antibody | Anti-ATP6V1H | Abcam | Cat# Ab187706 | WB (1:1000) |
| Antibody | Anti-COX1 | Self-made | PRAB2035 | WB (1:1000) |
| Antibody | Anti-cleaved caspase 3 | Cell signaling | RRID: AB_2070042 | WB (1:1000) |
| Antibody | Anti-cleaved PARP | Cell signaling | RRID: AB_10699459 | WB (1:1000) |
| Antibody | Anti-FTH1 | Cell signaling | RRID: AB_1903974 | WB (1:1000) |
| Antibody | Anti-Ferritin light chain | Abcam | RRID: AB_1523609 | WB (1:1000) |
| Antibody | Anti-GAPDH | Sigma-Aldrich | RRID: AB_796208 | WB (1:10000) |
| Antibody | Anti-GFAP | Abcam | RRID: AB_305808 | WB (1:1000) |
| Antibody | Anti-HA | Abcam | RRID: AB_301017 | WB (1:1000) |
| Antibody | Anti-HIF-1 alpha | Novus | RRID: AB_10000663 | WB (1:500) |
| Antibody | Anti-HPRT | Abcam | RRID: AB_297217 | WB (1:4000) |

*Continued on next page*

*Continued*

| Reagent type (species) or resource | Designation | Source or reference | Identifiers | Additional information |
|---|---|---|---|---|
| Antibody | Anti-LAMP1 | Sigma-Aldrich | RRID: AB_477157 | WB (1:1000) |
| Antibody | Anti-LC3B | Cell signaling | RRID: AB_2137707 | WB (1:1000) |
| Antibody | Anti-MBP | MPI-EM Goettingen | Klaus-Armin Nave | WB (1:1000) |
| Antibody | Anti-Total Rodent OXPHOS Cocktail | Abcam | RRID: AB_2629281 | WB (1:2000) |
| Antibody | Anti-PEX5 | Sigma-Aldrich | RRID: AB_10673419 | WB (1:1000) |
| Antibody | Anti-PLP | MPI-EM Goettingen | Klaus-Armin Nave | WB (1:1000) |
| Antibody | Anti-SQSTM1 | Abcam | RRID: AB_945626 | WB (1:8000) |
| Antibody | Anti-mtTFA | Abcam | RRID: AB_2651017 | WB (1:1000) |
| Antibody | Anti-TNFα | Abcam | Cat# Ab183218 | WB (1:1000) |
| Antibody | Anti-TOM20 | Proteintech Group | RRID: AB_2207530 | WB (1:1000) |
| Antibody | Anti-VAPB | Bethyl Laboratories | RRID: AB_2780841 | WB (1:1000) |
| Antibody | Anti-VDAC1 | Abcam | RRID: AB_443084 | WB (1:2000) |
| Antibody | Anti-VHL | Santa Cruz | RRID: AB_2215955 | WB (1:1000) |
| Antibody | Anti-mouse IgG | Jackson Laboratory | RRID: AB_2307392 | WB (1:4000) |
| Antibody | Anit-rabbit IgG | Jackson Laboratory | RRID: AB_2307391 | WB (1:4000) |
| Chemical compound, drug | Antimycin | Sigma-Aldrich | Cat# A8674 | 1 µM, 20 µM |
| Chemical compound, drug | Bafilomycin | Santa Cruz | Cat# sc-201550 | 500 nM, 50 nM |
| Chemical compound, drug | Bodipy[581/591] C11 | Thermo Fisher | Cat# D3861 | 1 µM |
| Chemical compound, drug | Deferoxamine mesylate | Sigma-Aldrich | Cat# D9533 | 300 µM |
| Chemical compound, drug | Dimethyl 2-oxoglutarate | Th. Geyer GmbH | Cat# 349631 | 5 mM |
| Chemical compound, drug | Dextran Oregon Green 488 | Thermo Fisher | Cat# D7170 | 0.4 mg/mL |
| Chemical compound, drug | Dextran Tetramethylrhodamine | Thermo Fisher | Cat# D1817 | 1 mg/mL |

Continued

| Reagent type (species) or resource | Designation | Source or reference | Identifiers | Additional information |
|---|---|---|---|---|
| Chemical compound, drug | Erastin | Cayman Chemical | Cat# 17754 | 10 µM |
| Chemical compound, drug | FCCP | Sigma-Aldrich | Cat# C2920 | 2 µM |
| Chemical compound, drug | Mito-FerroGreen | Dojindo Laboratories | Cat# M489 | 5 µM |
| Chemical compound, drug | FerroOrange | Goryo Chemical | Cat# GC904 | 1 µM |
| Chemical compound, drug | Ferrostatin | Sigma-Aldrich | Cat# SML0583 | 5 µM |
| Chemical compound, drug | $H_2$DCF-DA | Thermo Fisher | Cat# D399 | 10 µM |
| Chemical compound, drug | Ferric citrate | Sigma-Aldrich | Cat# F3388 | 150 µM |
| Chemical compound, drug | Lipofectamine RNAiMAX Reagent | Thermo Fisher | Cat# 13778–150 | |
| Chemical compound, drug | Lipofectamine 2000 | Thermo Fisher | Cat# 11668–019 | |
| Chemical compound, drug | MitoSox Red Mitochondrial Superoxide indicator | Thermo Fisher | Cat# M36008 | 1 µM |
| Chemical compound, drug | Necrostatin-1s | Abcam | Cat# Ab221984 | 5 µM |
| Chemical compound, drug | Oligomycin | Sigma-Aldrich | Cat# O4876 | 1 µM |
| Chemical compound, drug | Pierce anti-HA Magnetic Beads | Thermo Scientific | Cat# 88836 | 200 µL |
| Chemical compound, drug | Rotenone | Sigma-Aldrich | Cat# R8875 | 1 µM |
| Chemical compound, drug | Saliphenylhalamide | Omm Scientific Inc | Donald R. Stewart | 500 nM |
| Chemical compound, drug | Staurosporine | LC Laboratories | Cat# S-9300 | 1 µM |
| Chemical compound, drug | TriReagent | Sigma-Aldrich | Cat# T9424 | |
| Chemical compound, drug | z-VAD | InVivoGen | Cat# tlrl-vad | 20 µM |
| Commercial assay or kit | Iron assay Kit | Abcam | Cat# Ab83366 | |

*Continued*

| Reagent type (species) or resource | Designation | Source or reference | Identifiers | Additional information |
|---|---|---|---|---|
| Commercial assay or kit | Crystal RNA Mini Kit | BioLab Products | Cat# 31-010-404 | |
| Commercial assay or kit | Luna Universal qPCR Master Mix | New England Biolabs | Cat# M3003E | |
| Commercial assay or kit | iScript cDNA Synthesis Kit | Bio-Rad | Cat# 170–8891 | |
| Commercial assay or kit | Proteome Profiler Mouse Cytokine Array Kit, Panel A | R and D Systems | Cat# ARY006 | |
| Commercial assay or kit | Cell Titer-Glo Luminiscence Cell Viability Assay | Promega | Cat# G7572 | |
| Commercial assay or kit | Seahorse XFe Cell Mito Stress Test Kit | Agilent | Cat# 103015–100 | |
| Cell line (*M. musculsus*) | Murine embryonic fibroblast (wild type) | This paper | N/A | |
| Cell line (*M. musculsus*) | *Gaa*$^{-/-}$ murine embryonic fibroblasts | This paper | N/A | |
| Cell line (*M. musculsus*) | Primary cortical neurons | This paper | N/A | |
| Cell line (*M. musculsus*) | *Irp1*$^{-/-}$ and *Irp2*$^{-/-}$ murine embryonic fibroblasts | Israel Institute of technology | Esther Meyron-Holtz | |
| Cell line (*M. musculsus*) | Tmem192-3xHA murine embryonic fibroblasts | This paper | N/A | |
| Cell line (*M. musculsus*) | *Mcoln1*$^{-/-}$ murine embryonic fibroblasts | Ludwig Maximillians-University, Munich | Christian Grimm | |
| Genetic reagent (*M. musculus*) | Mouse: B6; 129-*Gaa*$^{tmRabn}$/J | Jackson Laboratory | JAX: 004154 | |
| Sequence-based reagent | qPCR primer sequences | Integrated DNA technologies | This paper (N/A) | Sequences in *Supplementary file 5* |
| Recombinant DNA reagent | Plasmid: pAcGFP1-Mito | Clontech | Cat# 632432 | |
| Recombinant DNA reagent | Plasmid: pLJC5-Tmem192-3xHA | Addgene | Cat# 102930 | |
| Transfected reagent (*M. musculus*) | siRNA to *Atp6v1h* | Integrated DNA technologies | This paper (N/A) | Sequences in *Supplementary file 5* |
| Transfected reagent (*M. musculus*) | siRNA to *Slc11a2* | Integrated DNA technologies | This paper (N/A) | Sequences in *Supplementary file 5* |
| Software, algorithm | Metascape | *Zhou et al., 2019* | http://metascape.org/gp/index.html#/main/step1 | |
| Software, algorithm | Prism version 8.0 | GraphPad Software Inc | https://www.graphpad.com/scientific-software/prism/ | |
| Software, algorithm | ImageJ version 1.51j8 | NIH | https://imagej.nih.gov/ij/ | |

*Continued on next page*

*Continued*

| Reagent type (species) or resource | Designation | Source or reference | Identifiers | Additional information |
|---|---|---|---|---|
| Software, algorithm | Ingenuity Pathway Analysis | Qiagen Bioinformatics | https://www.qiagenbioinformatics.com/products/ingenuity-pathway-analysis/ | |
| Software, algorithm | Partek Genomics Suite Analysis Software | Partek Inc | http://www.partek.com/partek-genomics-suite/ | |
| Software, algorithm | Wave Desktop software | Agilent | https://www.agilent.com/en/products/cell-analysis/software-download-for-wave-desktop | |
| Software, algorithm | Digigait Imaging and Analysis software | Mouse Specifics Inc | https://mousespecifics.com/digigait/ | |
| Software, algorithm | FlowJo v10 | Tree Star Inc | https://www.flowjo.com/solutions/flowjo/downloads | |

## Experimental model and subject details

### Mice

The *Gaa*$^{-/-}$ mice (B6:GaatmRbn/J) used in this study were purchased from the stock of The Jackson Laboratory, bred and housed under standard pathogen-free conditions at the animal facility of the European Neuroscience Institute with access to food and water ad libitum. All animal experiments were carried out in accordance with the European guidelines for animal welfare and were approved by the Lower Saxony Landesamt fur Verbraucherschutz and Lebensmittelsicherheit (LAVES) registration number 15–883. Except for iron-enriched chow experiments in which mixed sex cohorts were used at 2 months of age, all other experiments were performed on only male cohorts at 0, 2, 6 and 12 months of age. Regular chow contains 179 mg Fe/Kg, and the iron-enriched chow contains 500 mg Fe/Kg. The iron-enriched and control diet were purchased from Brogaarden (Denmark).

### Cell lines and primary cultures

Murine embryonic fibroblasts (MEFs) were prepared in-house from *Gaa*$^{-/-}$ mice and their wildtype littermate controls. MEFs from wild type, *Irp1*$^{-/-}$ and *Irp2*$^{-/-}$ mice were obtained from the Technion-Israel Institute of Technology (Esther Meyron Holtz). All these lines were cultured in Dulbecco's modified Eagle's medium with high glucose (Gibco) supplemented with 10% fetal bovine serum (FBS) and 1% Penicillin/Streptomycin at 37°C and 5% $CO_2$, in a humidified incubator, unless otherwise stated. Mycoplasma testing was routinely carried ou as described (*Murdoch et al., 2016*), and cell lines tested negative. Wild type C57BL/6J P0 pups were used to prepare primary cortical cultures as previously described (*Ferguson et al., 2007*). Briefly, mouse cortices were dissected under sterile conditions and digested in freshly prepared, warm enzyme solution (1.5 mM cysteine, 20 U/mL papain, 0.75 mM EDTA, 1.5 mM CaCl2 and 10 ul/mL DNase) in HBSS medium for 30 min at 37°C on a shaker. The enzyme solution was replaced with plating medium (Neurobasal medium supplemented with 5% FBS, 1% glutamax, 2% B27 supplement and 0.5% Penicillin/Streptomycin) for 5 min. Neuronal tissue was washed in HBSS medium and cultured in plating medium for 12 hr at 37°C and 5% $CO_2$, in a humidified incubator. The plating medium was replaced with neuronal medium (as plating medium without 5% FBS) and neurons were cultured under standard growth conditions (37°C and 5% $CO_2$) in a humidified incubator. All neuronal experiments were carried out with neurons at DIV 14. For transient siRNA-mediated silencing of Slc11a2 or Atp6v1h, MEFs were seeded and cultured for 24 hr prior to transfection under normal growth conditions. The medium was replaced with fresh growth medium. siRNA constructs along with scrambled non-targeting control and Lipofectamine RNAiMAX reagent were diluted separately in OptiMEM reduced serum medium according to supplier's instructions, mixed and incubated for five minutes at room temperature. The optimized

siRNA-Lipofectamine mix was added dropwise to cells, incubated for 72 hr, and harvested for subsequent assays.

## Generation of stable 3XHA-tagged Tmem192 cell line

Lentiviral suspensions were obtained by lipofectamine-mediated transfection of HEK293T packaging cells with an optimized mix of envelope and packaging plasmids, and pLJC5-Tmem192- 3xHA (a kind gift from David Sabatini, Addgene 102930). Following manufacturer's instructions, lentiviral vectors were concentrated using the Lenti-X concentrator. Mouse embryonic fibroblasts were transduced with concentrated viral suspension supplemented with 8 ug/ml Polybrene and cells with puromycin resistance were selected and polyclonally expanded. Overexpression efficiency was confirmed by anti-HA immunoblots of whole cells lysates.

## Behavior experiments

Mouse behavior experiments were performed as detailed in *Rostosky and Milosevic (2018)*. In brief, ventral plane videography for gait analysis was performed on $Gaa^{-/-}$ mice and their wild type littermate controls fed standard or iron-enriched chow for 2 months using Digigait instrumentation at a speed of $40 cms^{-1}$. For each group 13–19 male mice were used for behavior experiments. Results were analyzed with software from Mouse Specifics according to the manufacturer's protocol and differences were determined by the Welch's one-way ANOVA test with Dunnett's correction for multiple comparison to $Gaa^{-/-}$ mice fed standard diet.

## Cell culture drug treatments

All experiments involving drug treatments or iron supplementation in cells were carried out in MEFs prepared from wild type C57BL/6J mice under the standard growth conditions stated above (see 'Cell lines and primary cultures') unless otherwise stated. Briefly, following seeding and overnight incubation, cells were washed with warm PBS and treated for 24 hr with growth medium containing final concentrations of 0.5 µM bafilomycin, 0.5 µM saliphenylhalamide or 300 µM deferroxamine. Experiments involving iron supplementation were carried out for 48 hr with a final concentration 150 µM Fe-citrate (or Na-citrate as control) in growth medium. For each experiment, the used concentration of drugs is stated in the corresponding figure legends including dimethyl-ketoglutarate (5 mM), zVAD (20 µM), Ferrostatin (5 µM), Erastin (10 µM), Antimycin (20 µM) and Necrostatin-1s (5 µM).

## Cell viability and cell death assay

Cell viability or death was determined using the Cell Titer Glo Luminiscence cell viability assay (Promega) following manufacturer's instructions. Briefly 8000 fibroblasts per well were cultured in 200 uL of DMEM high glucose medium supplemented with 10% FBS and 1% Penicillin/Streptomycin as at least technical triplicates for each treatment in a 96-well plate. Following overnight culture, the initial number of cells prior to treatment was determined from five untreated wells. MEFs were then treated with the indicated drugs as described in each experimental condition for 24 hr or 48 hr in 100 uL of medium. Prior to cell number measurements, plates and reagents were equilibrated to room temperature. 100 uL of Cell Titer Glo reagent (Promega) was added to each well, mixed and the luminescence after 10 min of incubation was read in a microplate reader (Synergy H1, Biotek). Cell viability was determined from a standard calibration curve and reported as log2 fold change of cell numbers relative to that at the initial time point prior to treatment. Proliferation of cells is shown as positive log2 fold change while cell death is depicted as negative log2 fold change.

## Mouse brain sample preparation

For all biochemical and q-RT-PCR experiments, mouse tissues were harvested, snap frozen in liquid nitrogen and stored at −80˚C. Briefly, mice were anesthesized with isofluorane and euthanized by cervical dislocation. Mouse brain was dissected into cortices, hippocampi and rest of brain, rinsed in PBS, transferred into tubes and snap frozen in liquid nitrogen. These samples were frozen at −80˚C until processed for each experiment accordingly. For Immunohistochemical assays, mice were anethesized by peritoneal injection with 40 mg/mL chloral hydrate in PBS and perfused with 100 U/mL heparin followed by cold 4% *para*-formaldehyde (PFA) in PBS. Harvested brains were post-fixed overnight in 4%

PFA at 4°C, slowly dehydrated in 12–18% gradients of sucrose solutions and embedded in OCT (Tissue Tek) in a cryomold, frozen in isopentane set in liquid nitrogen and stored at −80°C.

## Fluorescence microscopy and image analysis

To determine cytoplasmic and mitochondrial labile iron levels by microscopy, MEFs were seeded in poly-L-lysine coated coverslips in 12-well plates, treated with bafilomycin for 24 hr and loaded with FerroOrange and Mito-FerroGreen respectively. Live cell imaging of cells was performed with a Nikon/Perkin Elmer Ultra VIEW VoX system (Perkin Elmer).

Mitochondrial morphology of bafilomycin-treated cells was assessed as described (*Fonseca et al., 2019*). Briefly, MEFs were electroporated (Amaxa Nucleofector) with pAcGFP1-Mito and seeded into poly-L-lysine coated coverslips in 12-well plates, treated for 24 hr with 0.5 uM bafilomycin and fixed with 3.7% PFA for 15 min at room temperature. About 0.2 µm thick Z-stacks were obtained with the 63x objective of the LSM800 Airyscan setup (Zeiss).

Frozen brain tissue as prepared before see 'Mouse brain sample preparation' above were used for Immunohistochemical assays to determine GFAP levels. In brief, 10 µM thick sagittal brain sections were cut with a cryostat, post-fixed and permeabilized with 0.1% Triton X-100. Immunofluorescence of the sections was performed as described (*Milosevic et al., 2011*). Image analysis was carried out as detailed in *Fonseca et al. (2019)* using ImageJ software version 1.51j8.

## Assessment of lysosomal acidification

Lysosomal pH was assessed as decribed (*Fernandez-Mosquera et al., 2019*). Briefly, we loaded MEFs with two dextran molecules, one labeled with Oregon Green (488) and the other with Tetramethylrhodamine (TMRM). The emission at 488 nm is quenched under acidic pH. The cells were incubated with dextrans in full medium during 6 hr, after which the dextrans were removed, the cells washed twice in PBS, and incubated in full medium overnight. The images were collected from live cells in imaging buffer in the next morning, using Spinning Disk confocal microscopy. The red channel was used to generate a mask on which the intensity of the green channel was measured. The intensity was then normalized to the area of the mask. WT fibroblasts treated with bafilomycin were used as positive control for loss of lysosomal acidification.

## Isolation of lysosomes

Lysosomes were purified by the LysoIP method as described in *Abu-Remaileh et al. (2017)*. Briefly 3xHA-tagged Tmem192 mouse embryonic fibroblasts were treated for 24 hr with or without 50 nM Baf. The medium was aspirated, and cells were washed twice with warm PBS, scraped into ice-cold KPBS (136 mM KCl, 10 mM KH2PO4, pH 7.25 adjusted with KOH) and centrifuged at 1000xg for 2 min at 4°C. 200 uL of anti-HA magnetic beads for each experimental condition were washed 3 times in 1 mL KPBS. The subsequent steps were carried out in the cold room. The cell pellets were resuspended in 1 mL KPBS, and 200 uL of the suspension was collected into separate tubes for whole cell extracts. The rest of the cell suspension was transferred into a 2 mL Dounce homogenizer and homogenized gently with 35 strokes while avoiding trapping air bubbles. The homogenate was centrifuged at 1000xg at 4°C for 2 min. The supernatant was carefully transferred into the tubes containing the pre-washed anti-HA magnetic beads and incubated for 10 min at 4°C on a rotator shaker. Immunoprecipitates were carefully washed three times in 1mLKPBS on a MagnaRack. Purified lysosomes were used for iron measurements or lysed for immunoblotting.

## Isolation of cortical mitochondria

Mouse cortices were harvested and mitochondria were purified from the tissues as described in *Lazarou et al. (2007)*. In brief, tissues were Dounce homogenized on ice in homogenization buffer (300 mM Trehalose, 10 mM KCl, 10 mM HEPES pH 7.4 supplemented freshly with 0.2% BSA and protease/phosphatase inhibitor cocktail), centrifuged twice at 700xg for 10 min at 4°C to pellet tissue debris and nuclear fractions. The supernatant was centrifuged at 11000xg for 15 min at 4°C to separate the mitochondria.

## Blue native (BN) –PAGE and complex IV activity staining

Isolated mitochondria were solubilized in buffer containing 1% digitonin and protein complexes were separated by BN-PAGE as previously described (Wittig and Schägger, 2005). For native complex IV activity staining, gels were equilibrated in 50 mM Potassium phosphate buffer (pH 7.4) and stained at 30°C with complex IV activity staining buffer (50 mM KPi pH 7.4, 0.5 mg/mL diamino-benzidine (DAB) and 1 mg/mL reduced cytochrome $c$) until the brown oxidized DAB stains were visible. The intensity of DAB stain (complex IV activity) was quantified with Imagequant TL (GE Healthcare Life Sciences). For native complex V activity staining, gels were equilibrated in 35 mM Tris, 220 mM Glycine (pH 8.3) and stained at 30°C in assay buffer (35 mM Tris, 220 mM glycine, 14 mM $MgSO_4$, 0.2% $Pb(NO_3)_2$, 8 mm ATP (pH 8.3)).

## Immunoblotting

Whole cell extracts of cultured MEFs or lysosomal extracts from purified lysosomes were lysed in 1.5% n-dodesylmaltoside (Roth) in PBS supplemented with protease and phosphatase inhibitor cocktail (Thermo Fisher Scientific) as described (Fernandez-Mosquera et al., 2019). For brain (cortical) lysates, snap frozen and ground tissues were lysed in RIPA buffer (50 mM Tris pH 8.0, 150 mM NaCl, 1% Triton X, 0.5% sodium deoxycholate and 0.1% SDS) freshly supplemented with protease and phosphatase inhibitor cocktail. Protein concentrations of whole cell extracts and tissue lysates were determined using a Bradford Protein assay (Bio-Rad). 50 μg of sample proteins per well were subjected to SDS-PAGE and transferred to polyvinylidene fluoride (PVDF) membranes (Amersham, Life Technologies). Membranes were blocked in 5% Milk in TBS tween and probed with the following primary antibodies: HIF-1 alpha (Novus), GAPDH (Sigma-Aldrich),SQSTM1 (Abcam), ATP6V1H (Abcam), VAPB (Bethyl), PEX5 (Sigma), anti-HA (Abcam), LAMP1 (Sigma Aldrich), (LC3B (Cell signaling), mtTFA (Abcam), Total rodent OXPHOS cocktail (Abcam), TNFα (Abcam), cleaved caspase 3 (Cell signaling), cleaved PARP (Cell signaling), GFAP (Abcam), PLP and MBP (Kind gift of K-A Nave, MPI-EM), FTH1 (Cell signaling), Ferritin light chain (Abcam), TOM20 (Proteintech), VDAC1 (Abcam), VHL (Santa Cruz) and self-made rabbit polyclonal antibodies to COX1 and ATP5B. HRP-conjugated secondary antibodies against mouse and rabbit IgGs (The Jackson Laboratory) were used. Band densitometric quantifications were determined using Image J software version 1.51j8.

## Mitochondrial respiration measurements

Mitochondrial oxygen consumption rate (OCR) was measured in fibroblasts using the XF96 Extracellular Flux analyzer (Seahorse Bioscience) as described (Yambire et al., 2019). Briefly, cells were seeded at 20000 cells per well in XF96 cell culture multi-well plates and cultured overnight under standard conditions stated previously (see 'Cell lines and primary cultures' above). Cells were then treated with the indicated drugs for 24 hr while XF96 cartridges were incubated overnight in XF calibrant at 37°C in a non-CO2 incubator. Prior to OCR measurements, the growth medium containing the indicated treatments of cells was exchanged with XF medium containing 1 μg/mL Hoechst dye, stained for 10 min to determine cell numbers, and subsequently incubated at 37°C in a non-CO2 incubator for 1 hr. For OCR profiles, 1 uM Oligomycin, 2 uM FCCP and 1 uM mix of Rotenone and Antimycin loaded into corresponding microwells in the XF96cartridge plate. Following equilibration of sensor cartridges, XF96 cell culture plate was loaded into the XF96 Extracellular Flux analyzer at 37°C and OCR was measured after cycles of mixing and acquiring data (basal) or inhibitor injection, mixing and data acquisition. Results were analyzed by the WAVE desktop software (Agilent) and normalized to the number of cells determined prior to assay.

## Quantitative RT-PCR

RNA from MEFs was extracted using the Crystal RNA mini Kit (Biolab,). For tissue RNA extraction, the TriReagent (Sigma, T9424) was used according to supplier instructions. RNA concentration and quality were determined using Nanodrop (PeqLab) and cDNA was synthesized with the iScript cDNA synthesis kit (Bio-Rad) following manufacturer's protocol. Each 9 μl reaction for q-RT-PCR was made of 4 μl diluted cDNA, 0.25 μl of each primer (from 25 μM stock) and 4.5 μl of Luna Universal Master Mix (New England Biolabs). The q-RT-PCR reactions were run on the QuantStudio 6 Flex Real-Time PCR system (Applied Biosystems). qPCR results were analyzed using the ΔΔCt method relative to the mean of housekeeping genes (Rps12, Hprt and Gapdh). Relative values for mtDNA and

nDNA genes were used to generate mtDNA copy number levels per nuclear genome. Each biological data point represents the average of at least technical triplicates.

## mtDNA copy number analysis

To measure relative mtDNA copy number levels, total DNA was isolated from cultured MEFs or tissues as described previously in *Cotney et al. (2007)*. Briefly, cells or tissues were lysed in 500 µL DNA extraction buffer (50 mM Tris-HCl pH 8.5, 0.25% SDS, 1 mM EDTA pH 8.0, 5 mM DTT) by boiling for 10 min. Samples were cooled to room temperature and incubated with 100 µg RNAse A for 3 hr at 37°C followed by incubation with 100 µg of Proteinase K at 55°C overnight. After Proteinase K digestion, samples were boiled at 95°C, allowed to cool to room temperature and the DNA concentration determined using Nanodrop (Peqlab). Diluted DNA samples were used along with mtDNA and nuclear DNA primers for qPCR as described above to determine relative mtDNA levels per genome.

## Flow cytometry

Measurements of mitochondrial superoxide levels, lipid peroxidation and cellular reactive oxygen species (ROS) levels were determined using MitoSox Red Superoxide indicator, Bodipy$^{581/591}$ C11 and H$_2$DCFDA (Thermo Fisher Scientific) respectively. Briefly, following treatments of cultured MEFs with indicated drugs for 24 hr, fibroblasts were washed in prewarmed PBS and loaded with either 1 µM MitoSox, 1 µM Bodipy$^{581/591}$ C11 or 10 µM H$_2$DCFDA during 30 min in the dark. Control cells were treated with H$_2$O$_2$ (5 mM) or Antimycin (20 µM) for 20 min as positive controls for increased levels of mitochondrial superoxides, lipid peroxidation or ROS. The mean fluorescent intensities corresponding to steady state levels of mitochondrial superoxides (Mitosox), lipid peroxidation (Bodipy$^{581/591}$ C11) or ROS levels (H$_2$DCFDA) were determined by flow cytometry (FACs Canto II, BD Biosciences). Data were collected from 20,000 cells and results were analyzed by FlowJo vX.0.7 (Tree Star Inc).

## Measurement of cellular or lysosomal iron levels

Determination of cellular total, ferric and ferrous iron levels were carried out using the Iron assay Kit (Abcam) according to manufacturer's instructions. Previously cultured and bafilomycin-treated MEFs were washed in prewarmed PBS, scraped into ice-cold PBS and pelleted at 1000xg for 5 min. Cell pellets or purified lysosomes were homogenized in assay buffer on ice using a Dounce homogenizer, centrifuged at 16000xg for 10 min and the supernatant collected for the iron assay. 25 µL of samples were made up to 100 µL in a 96-well plate with assay buffer, and incubated for 30 min at 37°C with 5 µL iron reducer (for total iron) or assay buffer (for ferrous iron) along with standards. 100 µL of iron probe was added to each reaction, mixed and incubated for a further 1 hr at 37°C in the dark. The absorbance at 593 nm, which corresponds to the iron levels, was determined using a microplate reader (Synergy H1, Biotek). Iron concentrations were calculated from the standard curve and normalized to the amount of protein for each sample, determined by the Bradford protein assay. The average of technical triplicates was used for each biological sample.

## Magnetic resonance spectroscopy (MRS) and magnetic resonance imaging (MRI)

Mice underwent MRS, T$_1$, T$_2$, and MTR measurements of the cerebral cortex, striatum, and thalamus at 37°C. After induction of anaesthesia by intraperitoneal injection of ketamine (100 mg/kg b.w.) and xylazine (10 mg/kg b.w.), animals were intubated with a purpose-built polyethylene endotracheal tube (0.58 mm inner diameter, 0.96 mm outer diameter) and artificially ventilated using an animal respirator (TSE, Bad Homberg, Germany) with a respiratory rate of 25 breaths per minute and an estimated tidal volume of 0.35 ml under 0.5% isoflurane as previously described (*Watanabe et al., 2016*). The animals were then placed in a prone position on a purpose-built palate holder equipped with an adjustable nose cone. Respiratory movement of the abdomen as well as rectal temperature were monitored by a unit supplied by the manufacturer (Bruker Biospin MRI GmbH, Ettlingen, Germany). At 9.4 T (Bruker Biospin MRI GmbH, Ettlingen, Germany), localized proton MRS (STEAM, TR/TE/TM = 6000/10/10 ms) was performed with the use of a birdcage resonator (inner diameter 70 mm) and a saddle-shaped quadrature surface coil (both Bruker Biospin MRI GmbH, Ettlingen,

Germany) on anesthetized mice as previously described (*Watanabe et al., 2016*). Shimming of the $B_0$ field was carried out by FASTMAP (*Gruetter, 1993*). A (20 mm)$^3$ volume-of-interest was centered on the right striatum (squares in *Figure 4—figure supplement 1D*). Water saturation was accomplished by means of three Gaussian-shaped CHESS radiofrequency (RF) pulses (90°−90°−180°), each of which with a duration of 7.83 ms and a bandwidth of 350 Hz. Overall duration of the CHESS module was 147 ms. Each CHESS pulse was followed by an associated spoiler gradient and a 37 ms outer volume saturation module covering a range of 3 mm around the volume-of-interest without gap. Metabolite quantification involved spectral evaluation by LCModel (*Provencher, 1993*) and calibration with brain water concentration of 79% (*Duarte et al., 2014*), for which the unsuppressed water proton signal served as internal reference. Metabolites with Cramer-Rao lower bounds above 20% were excluded from further analysis. Relaxation times and magnetization transfer ratios are determined in selected regions of the brain. $T_2$ relaxation times of water protons were determined by a multi-echo spin-echo MRI (TR/TE = 2500/10–123 ms). $T_1$ relaxation times were determined with the use of a spin-echo saturation recovery sequence and 7 TR values from 0.15 to 6 s. For measurements of the magnetization transfer ratio, an off-resonance Gaussian-shaped RF pulse with a frequency offset of −3.6 kHz, duration of 8.5 ms, and a flip angle of 128° was incorporated into a spin density-weighted gradient-echo MRI sequence (RF-spoiled 3D FLASH, TR/TE = 19/4.2 ms, flip angle 5°, measuring time 97 s) at a resolution of 200 × 200 × 400 μm$^3$. The duration and power of the off-resonance pulse was optimized to observe the transfer of saturation from non-water protons to water protons in brain of mice in vivo (*Natt et al., 2003*; *Watanabe et al., 2012*). For the evaluation of MRI signal intensities, square-shaped regions of interest with were selected in a standardized manner in the cerebral cortex (98 pixels), the striatum (100 pixels), and the thalamus (61 pixels) (*Figure 4—figure supplement 1D*). The analysis followed a strategy previously developed for intraindividual comparisons of MR images obtained after manganese administration (*Watanabe et al., 2004*). Volumetric assessments were obtained by analyzing proton-density-weighted images (RF-spoiled 3D FLASH, TR/TE = 22/7.7 ms, flip angle 10°, fat suppression 90°, 117 μm isotropic resolution, measuring time 12 min, *Figure 4—figure supplement 1D*) using software provided by the manufacturer (Paravision 5.0, Bruker Biospin, Ettlingen, Germany). After manually outlining the whole brain and the ventricular spaces in individual sections, respective areas were calculated (in mm$^2$), summed and multiplied by the section thickness. Significant differences between two groups of data were determined by Mann-Whitney´s U-test.

## Cytokine measurements

Brain tissue cytokine levels were measured using the Mouse cytokine array Panel A (R and D Systems) following protocols described by the manufacturer. Briefly cortices from $Gaa^{-/-}$ mice mice and their wild type littermate controls, fed iron-enriched or standard chow for 2 months were harvested, snap frozen in liquid nitrogen and ground using a tissue grinder. Tissues were lysed in RIPA buffer and spun at 10000xg for 5 min. The supernatants were collected and their protein concentrations determined by the BCA protein assay (Thermo Fisher Scientific). Nitrocellulose membranes containing capture antibodies were blocked in array buffer 6 for 1 hr. Samples were prepared by diluting 1 mg of tissue lysates in 500 μL array buffer 4 containing 15 μL Detection antibody cocktail and made up to 1.5 mL with array buffer 6. Block buffer on membranes was replaced with prepared samples. Following overnight incubation at 40C, membranes were washed three times in wash buffer for 10 min each, incubated for 30 min with Strepatavidin-HRP on a rocking platform, washed three more times and incubated for 1 min with the Chemi Reagent mix. Membranes were developed onto X-ray films by a chemiluminiscence developer (AGFA) and the pixel density (using ImageJ) of each spot representing the level of the corresponding cytokine determined and normalized to that of the average of reference spots.

## Quantification and statistical analysis

### Statistical analysis

All statistical analyses were carried out using Prism version 8.0 software. Unless otherwise stated in figure legends, to compare means between two groups, a two-tailed unpaired *t*-test with Welch's correction was used for normally distributed data. Non-parametric, two-group means were compared by a two-tailed Mann-Whitney test. Welch's one-way ANOVA, followed by Dunnett's test for

multiple comparisons was performed for multi-group (at least three) comparisons. For data on cytokine array experiments, the Brown-Forsythe's one-way ANOVA test was used given the identical values in at least one of the coconditions. Measures were summarized as graphs displaying mean ± SEM, of at least three independent biological replicates. Means of control samples are typically centered at one (or 100%) to ensure easier comparisons unless otherwise noted. Estimated p values are either stated as the actual values or denoted by $*p<0.05$; $**p<0.01$; $***p<0.001$. Differences were only considered as statistically significant when the p value was less than 0.05.

## Data and software availability

The microarray and RNA sequencing data analysis was performed using Partek Software Suite. Microarray data analysis was performed as described in *Raimundo et al. (2009)*, using RMA normalization. RNAseq data was aligned to the reference genome mm10 by the BowTie algorithm, and the transcripts were quantified using the 'mm10 - Ensembl Transcripts release 95' as reference. For differential expression analysis, Bonferroni multi-test correction was applied, and adjusted p-value<0.05 was considered significant. Ingenuity Pathway Analysis was used for assessment of transcription factor activity, as described in *Yambire et al. (2019)*. The RNAseq data generated in this study from the *Gaa*-KO and WT cortices is deposited in Gene Expression Omnibus under the serial number GSE134704.

# Acknowledgements

This work was supported by a Starting Grant from the European Research Council (337327, MitoPex-Lyso), to NR. This project stems from results obtained under the support of a research grant from the Acid Maltase Deficiency Association (to NR). IM lab was supported by a Emmy Noether award from the Deutsche Forschungsgemeinschaft (DFG). NR and IM were supported by the Collaborative Research Center SFB1190-P02 by DFG.

The authors thank the technical support of Dirk Schwitters, and thank Adrian M Raimundo for discussions on the project. We thank Prof. Dörthe Katschinski (Universitätsmedizin Göttingen) for the primers to test HIF targets, and Prof. David Sabatini for the Addgene construct 102930.

# Additional information

## Funding

| Funder | Grant reference number | Author |
| --- | --- | --- |
| H2020 European Research Council | 337327 | Nuno Raimundo |
| Acid Maltase Deficiency Association | | Nuno Raimundo |
| Deutsche Forschungsgemeinschaft | Emmy Noether award | Ira Milosevic |
| Deutsche Forschungsgemeinschaft | SFB1190-P02 | Ira Milosevic<br>Nuno Raimundo |

The funders had no role in study design, data collection and interpretation, or the decision to submit the work for publication.

## Author contributions

King Faisal Yambire, Conceptualization, Formal analysis, Validation, Investigation; Christine Rostosky, Takashi Watanabe, David Pacheu-Grau, Sylvia Torres-Odio, Angela Sanchez-Guerrero, Ola Senderovich, Formal analysis, Investigation; Esther G Meyron-Holtz, Conceptualization, Resources, Formal analysis, Supervision; Ira Milosevic, Resources, Formal analysis, Supervision, Investigation; Jens Frahm, Formal analysis, Supervision; A Phillip West, Conceptualization, Resources, Formal analysis, Supervision, Investigation; Nuno Raimundo, Conceptualization, Resources, Data curation, Formal analysis, Supervision, Funding acquisition, Validation, Investigation

### Author ORCIDs
Ira Milosevic ⬤ http://orcid.org/0000-0001-6440-3763
Jens Frahm ⬤ http://orcid.org/0000-0002-8279-884X
Nuno Raimundo ⬤ https://orcid.org/0000-0002-5988-9129

### Ethics
Animal experimentation: The experiments were performed under the permit 15-883 by the authority for animal research in Lower Saxony, Germany (LAVES).

### Decision letter and Author response
Decision letter https://doi.org/10.7554/eLife.51031.sa1
Author response https://doi.org/10.7554/eLife.51031.sa2

# Additional files
### Supplementary files
• Supplementary file 1. Annotation of disease or functions (Ingenuity Pathway Analysis) for the category of cell death and survival, for the differentially-expressed gene list obtained from dataset GSE47836.

• Supplementary file 2. Annotation of disease or functions (Ingenuity Pathway Analysis) for the category of cell death and survival, for the differentially-expressed gene list obtained from dataset GSE60570.

• Supplementary file 3. Annotation of disease or functions (Ingenuity Pathway Analysis) for the category of cell death and survival, for the differentially-expressed gene list obtained from dataset GSE16870.

• Supplementary file 4. Concentration and concentration ratios of metabolites determined by in vivo MRS.

• Supplementary file 5. List of qPCR primers and siRNA sequences for mouse genes.

• Supplementary file 6. Antibody validation.

• Transparent reporting form

### Data availability
We generated RNAseq data from brain of mice (WT and KO), which is deposited in Gene Expression Omnibus under the serial number Series GSE134704.

The following dataset was generated:

| Author(s) | Year | Dataset title | Dataset URL | Database and Identifier |
|---|---|---|---|---|
| Yambire KF, Raimundo N | 2019 | Cortical transcriptome reveals widespread inflammation in brain of Gaa-/- mice | https://www.ncbi.nlm.nih.gov/geo/query/acc.cgi?acc=GSE134704 | NCBI Gene Expression Omnibus, GSE134704 |

The following previously published datasets were used:

| Author(s) | Year | Dataset title | Dataset URL | Database and Identifier |
|---|---|---|---|---|
| Straud S, Roth MG | 2008 | HeLa cells treated with V-ATPase inhibitors or with desoxyferramine compared to HeLa in DMSO or medium with low LDL | https://www.ncbi.nlm.nih.gov/geo/query/acc.cgi?acc=GSE16870 | NCBI Gene Expression Omnibus, GSE16870 |
| Stingele S, Stoehr G, Dürrbaum M, Storchová Z | 2013 | Human colon carcinoma cell line treated with bafilomycin A1 | https://www.ncbi.nlm.nih.gov/geo/query/acc.cgi?acc=GSE47836 | NCBI Gene Expression Omnibus, GSE47836 |
| Santaguida S, Vazile E, White E, Amon A | 2014 | Aneuploidy-induced cellular stresses limit autophagic degradation. | https://www.ncbi.nlm.nih.gov/geo/query/acc.cgi?acc=GSE60570 | NCBI Gene Expression Omnibus, GSE60570 |

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
