## [Decision Letter]

**Acceptance summary:**

There is broad interest in diseases related to loss of lysosome acidification, and your studies on the link between this process and cellular iron regulation have important implications for our understanding of the etiology of a number of disease processes and the links between lysosome and mitochondrial function and inflammatory signaling.

**Decision letter after peer review:**

Thank you for submitting your article "Impaired lysosomal acidification triggers iron deficiency, necrotic cell death and inflammation in vivo" for consideration by *eLife*. Your article has been reviewed by three peer reviewers, and the evaluation has been overseen by Suzanne Pfeffer as the Reviewing and Senior Editor. The following individuals involved in review of your submission have agreed to reveal their identity: Peter S McPherson (Reviewer #1).

The reviewers have discussed the reviews with one another and the Reviewing Editor has drafted this decision to help you prepare a revised submission.

In this manuscript Raimundo and colleagues investigate how inhibition of lysosomal acidification impacts cellular iron homeostasis, inflammation and cell death. The authors pursue mechanistic studies in cultured cells, before moving into an in vivo model of lysosome dysfunction, and develop a model that links lysosome de-acidification to cytosolic iron deficiency, inflammatory gene expression and motor dysfunction. This is a well-written manuscript that presents a number of interesting findings in a clear and logical manner.

Before publication can be recommended, the following issues need to be addressed to ensure that the small molecule reagents you have employed are consistent with the conclusions of your study.

1) Bafilomycin A is used to study loss of lysosomal acidification. To ensure that this is the case (as opposed to an unrelated effect of v-ATPase inhibition), compounds that dissipate or clamp lysosomal pH in an vATPase-independent manner (i.e. chloroquine) should be tested; Also, there are some concerns about the concentration of Baf used. 500nM is much higher than the IC50, which is in the low nanomolar. To rule out indirect effects, it would be important to carry out RNAi of an essential vATPase subunit and determine whether it also results in activation of the hypoxic response and disrupted mitochondrial function.

2) The authors imply that v-ATPase inhibition results in build-up of iron in the lysosome. Whole-cell measurements of iron levels show a decrease in both Fe^2+^ and Fe^3+^ following Baf treatment (Figure 1E). However, if iron is trapped in lysosomes, the expectation is that its subcellular distribution should be altered, not its total concentrations. The authors could repeat the iron measurements described in the Materials and methods after fractionating the cells to separate light organelle fraction from cytosol (an easy procedure that should confirm reduced cytosolic iron levels), or upon lysosomal immunoprecipitation (i.e. PMID: 29074583). If iron accumulation within lysosomes is the ultimate cause of the observed mitochondrial dysfunction, this phenotype should be recapitulated by ablation of the main iron-exporting permeases, SLC11A2 and MCOLN1, either individually or in combination.

3) An alternative explanation for the observed mitochondrial dysfunction in Baf-treated cells is impairment of mitophagy. Are Baf-dependent induction of the hypoxic response and mitochondrial dysfunction identical between wild-type cells and cells lacking essential autophagic factors (i.e. Atg7)?

4) The authors use a RIPK-1 inhibitor, known as necrostatin, to imply a role for necroptosis in Baf-induced cell death. It should be noted that the specificity of action of necrostatin is questionable (i.e. https://doi.org/10.1038/cddis.2012.176), thus it is highly recommended that the authors further support this conclusion with RNAi-based experiments. Demonstration that the cell death process is necroptosis may be beyond the scope of the present story and could be removed.

5) The GAA-defective cells and mice are assumed, but not shown, to have defective lysosomal acidification. Given the importance of this assumption for their model, the authors should re-check it on their own at least in the cell model. Also, GAA loss has more ample consequences than just loss of lysosomal acidity. This caveat should be acknowledged and adequately discussed.

---

## [Author Response]

[…]1) Bafilomycin is used to study loss of lysosomal acidification. To ensure that this is the case (as opposed to an unrelated effect of v-ATPase inhibition), compounds that dissipate or clamp lysosomal pH in an vATPase-independent manner (i.e. chloroquine) should be tested; Also, there are some concerns about the concentration of Baf used. 500nM is much higher than the IC50, which is in the low nanomolar. To rule out indirect effects, it would be important to carry out RNAi of an essential vATPase subunit and determine whether it also results in activation of the hypoxic response and disrupted mitochondrial function.

We obtained similar results with lower concentrations (50nM) of bafilomycin and chloroquine (added as Figure 2—figure supplement 1C, Figure 3—figure supplement 1B, E and G). Furthermore, we also silenced ATP6V1H (Figure 2—figure supplement 1E) and obtained similar results (Figure 2—figure supplement 1E-H, and Figure 3—figure supplement 2A-C).

2) The authors imply that v-ATPase inhibition results in buildup of iron in the lysosome. Whole-cell measurements of iron levels show a decrease in both Fe^2+^ and Fe^3+^ following Baf treatment (Figure 1E). However, if iron is trapped in lysosomes, the expectation is that its subcellular distribution should be altered, not its total concentrations. The authors could repeat the iron measurements described in the Materials and methods after fractionating the cells to separate light organelle fraction from cytosol (an easy procedure that should confirm reduced cytosolic iron levels), or upon lysosomal immunoprecipitation (i.e. PMID: 29074583). If iron accumulation within lysosomes is the ultimate cause of the observed mitochondrial dysfunction, this phenotype should be recapitulated by ablation of the main iron-exporting permeases, SLC11A2 and MCOLN1, either individually or in combination.

We isolated lysosomes using the method suggested by the reviewers, and obtained a clean lysosomal preparation without endoplasmic reticulum or mitochondrial contaminants (new Figure1H). In agreement with the rest of our data, we observed an increase in total Fe levels in the lysosome following treatment with 50nM bafilomycin (new Figure 1G). We tested *Mcoln1*-KO fibroblasts, in which we silenced *Slc11a2* (DMT1), to affect both lysosomal iron exporters simultaneously. We observed decreased ferritin light chain protein levels and increased transferrin receptor (supporting the iron deficiency phenotype), decreased O_2_ consumption and decreased OXPHOS subunits protein and transcript levels, (Figure 3—figure supplement 2D-H).

3) An alternative explanation for the observed mitochondrial dysfunction in Baf-treated cells is impairment of mitophagy. Are Baf-dependent induction of the hypoxic response and mitochondrial dysfunction identical between wild-type cells and cells lacking essential autophagic factors (i.e. Atg7)?

While it is expectable that vATPase inhibition impacts the autophagic pathway, and therefore mitophagy, the decrease in mitochondrial protein levels suggests that mitophagy is not the primary driver of the phenotype, which seems rather a consequence of the inhibition of the transcriptional program of mitochondrial biogenesis (even when the transcriptional program of mitochondrial biogenesis is down-regulated but mitophagy is impaired, we can observe a strong accumulation of mitochondrial proteins and mass, as described in our recent study Yambire et al., 2019). This is further supported by the rescue of mitochondrial mass and function when the cell medium is supplemented with Fe-citrate simultaneously with bafilomycin or Saliphe treatments without affecting levels of SQSTM1 and LC3B.

4) The authors use a RIPK-1 inhibitor, known as necrostatin, to imply a role for necroptosis in Baf-induced cell death. It should be noted that the specificity of action of necrostatin is questionable (i.e. https://www.nature.com/articles/cddis2012176), thus it is highly recommended that the authors further support this conclusion with RNAi-based experiments. Demonstration that the cell death process is necroptosis may be beyond the scope of the present story and could be removed.

We tested the more stable Nec1s. At the suggested concentrations for Nec1s, we didn’t find inhibition of bafilomycin- or saliphe-induced cell death (Figure 4—figure supplement 1E). We agree with the reviewers that detailing the exact mechanism of cell death in this context is not absolutely essential and will deviate from the main message of this study. Therefore, we reworded the text, and refer to the cell death process as non-apoptotic, and moved most of these data to a supplementary figure.

5) The GAA-defective cells and mice are assumed, but not shown, to have defective lysosomal acidification. Given the importance of this assumption for their model, the authors should re-check it on their own at least in the cell model. Also, GAA loss has more ample consequences than just loss of lysosomal acidity. This caveat should be acknowledged and adequately discussed.

We have tested the lysosomal pH in the *Gaa*^-/-^ fibroblasts, and observed decreased acidification (new Figure 2F-G), in agreement with the findings from Nina Raben and colleagues(Fukuda et al., 2006). We completely agree that the loss of GAA has broader consequences besides lysosomal acidification, and modified the text accordingly. Nevertheless, the supplementation of Fe also has a strong beneficial effect on the *Gaa*^-/-^ cells and *Gaa*^-/-^ mice, suggesting that the effect of impaired lysosomal acidification in Fe homeostasis is a significant component of the response to GAA loss. We have also included the assessment of lysosomal pH when *Atp6v1h* is silenced, which was expectably less acidic (Figure 2—figure supplement 1).